# Membrane curvature induced by proximity of anionic phospholipids can initiate endocytosis

Takashi Hirama[1,2,3], Stella M. Lu[1,3,4], Jason G. Kay[5], Masashi Maekawa[3,6], Michael M. Kozlov[7], Sergio Grinstein[1,4,8] & Gregory D. Fairn [3,4,8,9]

The plasma membrane is uniquely enriched in phosphatidylserine (PtdSer). This anionic phospholipid is restricted almost exclusively to the inner leaflet of the plasmalemma. Because of their high density, the headgroups of anionic lipids experience electrostatic repulsion that, being exerted asymmetrically, is predicted to favor membrane curvature. We demonstrate that cholesterol limits this repulsion and tendency to curve. Removal of cholesterol or insertion of excess PtdSer increases the charge density of the inner leaflet, generating foci of enhanced charge and curvature where endophilin and synaptojanin are recruited. From these sites emerge tubules that undergo fragmentation, resulting in marked endocytosis of PtdSer. Shielding or reduction of the surface charge or imposition of outward membrane tension minimized invagination and PtdSer endocytosis. We propose that cholesterol associates with PtdSer to form nanodomains where the headgroups of PtdSer are maintained sufficiently separated to limit spontaneous curvature while sheltering the hydrophobic sterol from the aqueous medium.

[1] Program in Cell Biology, The Hospital for Sick Children, 555 University Avenue, Toronto, ON, Canada M5G 1X8. [2] Department of Respiratory Medicine, Saitama Medical University, Moroyama, Saitama 3500495, Japan. [3] Keenan Research Centre for Biomedical Science, St. Michael's Hospital, 209 Victoria Street, Toronto, ON, Canada M5B 1T8. [4] Department of Biochemistry, University of Toronto, Toronto, ON, Canada M5S 1A8. [5] Department of Oral Biology, School of Dental Medicine, University at Buffalo, Buffalo, NY 14214, USA. [6] Department of Biochemistry and Molecular Genetics, Ehime University Graduate School of Medicine; Division of Cell Growth and Tumour Regulation, Proteo-Science Center, Ehime University, Toon, Ehime 7910295, Japan. [7] Department of Physiology and Pharmacology, Room 546, Sackler Faculty of Medicine, Tel Aviv University, Tel Aviv 69978, Israel. [8] Institute of Medical Science, Faculty of Medicine, University of Toronto, Toronto, ON, Canada M5S 1A8. [9] Department of Surgery, University of Toronto, Toronto, ON, Canada M5T 1P5. Takashi Hirama and Stella M. Lu contributed equally to this work. Correspondence and requests for materials should be addressed to G.D.F. (email: fairng@smh.ca)

The membranes that define the boundaries of individual cellular compartments differ in composition; the concentration of cholesterol and individual phospholipids can vary widely between organelles. A striking example is provided by phosphatidylserine (PtdSer), which is many-fold more abundant in the plasma membrane (PM) than in the endoplasmic reticulum (ER), where it is synthesized. PtdSer is restricted to the inner leaflet of the plasmalemmal bilayer, and alterations in this asymmetric distribution signal the clearance of apoptotic cells and are key to effective blood clotting. A growing body of literature suggests that PtdSer is transported directly from the ER to the PM by members of the oxysterol-binding protein family[1, 2]. Because PtdSer is present on secretory vesicles, delivery by vesicular transport also contributes to its abundance in the PM[3–5].

Like PtdSer, cholesterol is also enriched in the PM compared to the ER[6] and is also believed to be asymmetrically distributed across the plasmalemmal bilayer[7]. Interestingly, recent work has demonstrated that the proper transbilayer distribution of cholesterol relies on PtdSer, especially, PtdSer (18:0/18:1)[8]. In cells with reduced PtdSer content, or in those where PtdSer relocalized to endomembranes, more cholesterol is observed in both the endocytic pathway and in the exofacial leaflet of the plasma membrane[8, 9]. These results suggest that association of cholesterol and PtdSer in the inner leaflet of the PM is critical for cholesterol retention and proper transbilayer distribution.

It remains unclear whether cholesterol plays a complementary role in dictating the distribution of PtdSer. To investigate this possibility, we manipulated the content of plasmalemmal cholesterol and monitored the distribution of PtdSer using genetically-encoded biosensors and biochemical methods. We find that upon the rapid removal of cholesterol or the increase in the density of PtdSer that there is a surge in spontaneous membrane curvature that facilitates endocytosis.

## Results

**Redistribution of PtdSer upon rapid cholesterol extraction.** To determine if cholesterol is required to maintain the plasmalemmal pool of PtdSer in mammalian cells, cholesterol was extracted from the PM using methyl-β-cyclodextrin (mβCD)[9]. Due to the high rate of spontaneous flip-flop of cholesterol across bilayers, mβCD effectively depletes cholesterol from both leaflets of the plasmalemma[10]. The distribution of PtdSer was monitored using the C2 domain of lactadherin (LactC2) fused to either GFP or mCherry[11]. Acute removal of cholesterol with mβCD caused a marked depletion of plasmalemmal LactC2, accompanied by a substantial redistribution of the probe to internal structures (Fig. 1a, b). It is conceivable that the ability of the biosensor to recognize PtdSer is affected by cholesterol. We used in vitro FRET-based measurements to assess this possibility. As illustrated in Fig. 1c, the association of recombinant LactC2 with PtdSer in liposomes was unaffected by cholesterol; the binding affinity of the probe was indistinguishable whether cholesterol was present or omitted.

The detachment of the LactC2 probe from the PM may be indicative of either relocalization of PtdSer, or of its degradation. We quantified the total cellular content of PtdSer by biochemical means to assess whether degradation had occurred, and found no significant difference between control and mβCD-treated cells (Fig. 1d), ruling out the removal or metabolic conversion of PtdSer to other phospholipid species. As LactC2 is expressed by the cells following transfection, it detects only PtdSer on the cytosolic leaflet of cellular membranes. The detachment of plasmalemmal LactC2 could, therefore, result from scrambling or flopping of PtdSer to the outer leaflet of the PM. However, unlike what is observed when scrambling of PtdSer is induced by

calcium ionophores (Fig. 1e), annexin V failed to detect exofacial PtdSer in cells treated with mβCD, ruling out the occurrence of scrambling or flopping. Together, the preceding observations strongly suggested that depletion of cholesterol caused a loss of PtdSer from the PM, attributable to redistribution to internal membranes.

To validate that a net loss of plasmalemmal PtdSer occurred in response to cholesterol depletion, we quantified the content of PtdSer of membranes isolated before and after treatment with mβCD. For this analysis, suspended cells were allowed to adhere to polycationic beads and subsequently sheared off, leaving the plasma membrane on the bead surface[12] (Supplementary Fig. 1). Extraction and analysis of the lipids of the adherent membranes revealed that cholesterol depletion caused a ≈50% reduction of the plasmalemmal PtdSer (Fig. 1f). The loss of PtdSer detected biochemically is consistent with and can account for the detachment of the LactC2 biosensor.

Next, we sought to determine the destination of the PtdSer lost from the PM. This was accomplished by quantifying the amount of GFP-LactC2 associated with well-established organellar markers (Fig. 1g, Supplementary Fig. 2). This analysis demonstrated that endocytic membranes including early, recycling and late endosomes/lysosomes became enriched in PtdSer after cholesterol removal. In contrast, the ER, Golgi complex, and mitochondria did not display any appreciable increase in LactC2 fluorescence. Thus, our results are indicative of PtdSer relocalization to the endocytic compartment upon loss of plasmalemmal cholesterol.

**PtdSer internalization is ATP independent.** The mechanism underlying PtdSer redistribution to endocytic compartments was investigated next. As canonical endocytic processes require energy, we tested whether ATP is necessary for the redistribution. Cellular ATP was acutely depleted by the addition of antimycin and 2-deoxy-D-glucose, a procedure that by itself did not noticeably alter LactC2 distribution (Fig. 2a). This was followed by cholesterol extraction and re-assessment of the distribution of PtdSer. Unlike cells with normal ATP levels, in ATP-depleted cells, LactC2 did not relocalize to membranes in the juxtanuclear region following treatment with mβCD. Instead, LactC2 consistently appeared as thick and irregular strands at or near the PM (Fig. 2a). We reasoned that perhaps PtdSer-enriched membranes had in fact internalized but, due to the absence of ATP, were not transported away from the PM. To determine if the LactC2-positive structures observed in ATP-depleted cells were indeed internalized or instead were continuous with the PM we then added the membrane-impermeant dye, FM4-64. As shown in Fig. 2a, FM4-64 exquisitely delineated the PM in ATP-depleted cells, vouching for their integrity. Importantly, the LactC2 strands detected near the PM failed to stain with FM4-64, implying that they are adjacent, but unconnected to the plasmalemma. These results suggest that relocalization of PtdSer to endomembranes following cholesterol extraction consists of two steps: an initial ATP-independent internalization event, followed by a centripetal displacement that requires ATP.

While being ATP independent, the redistribution of PtdSer is nevertheless temperature sensitive. We ensured that mβCD properly extracted cholesterol at 4 °C (Fig. 2b) and monitored the distribution of LactC2. Under these conditions, PtdSer failed to internalize, since LactC2 remained at the cell surface, where it colocalized with the surface marker, FM4-64 (Fig. 2c).

**Removal of cholesterol increases surface charge density.** The above observations led us to postulate that the process driving internalization was not metabolic in nature. As cholesterol

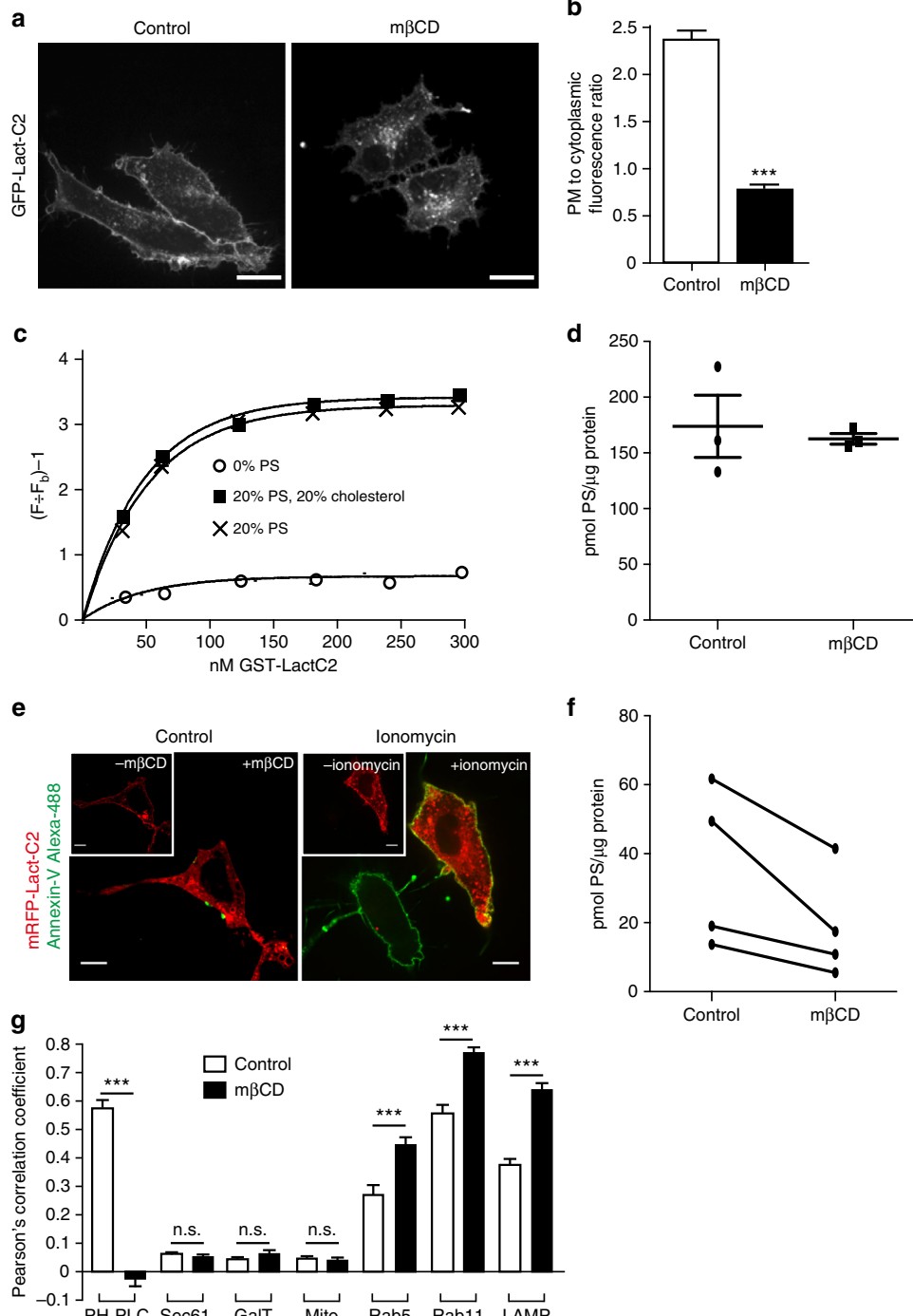

**Fig. 1** PtdSer is redistributed upon cholesterol extraction. **a** Distribution of GFP-LactC2 in HeLa cells that were untreated or treated with 10 mM mβCD for 30 min. Images in A are representative of at least 4 experiments of each type. **b** Quantitation of the ratio of PM to cytoplasmic GFP-LactC2 from A, $n = 40$ cells. **c** Quantitation of GST-LactC2 to liposomes containing egg-PtdCho, dansyl-PtdEtn, 0% or 20% PtdSer, with 0% or 20% cholesterol. The term $(F \div Fb) - 1$ refers to the observed FRET resulting from the proximity of dansyl-PtdEtn in the liposome to tryptophan residues in GST-LactC2. **d** Comparison of the total PtdSer content of control and mβCD-extracted cells. PtdSer extracted from cells was reacted with fluorescamine and quantified by measuring fluorescence intensity after separation by TLC, then normalized to total protein of the cells. Data are means ± s.e.m from three separate experiments. **e** HeLa cells expressing mRFP-LactC2 (red) were treated with either 10 mM mβCD (left) or 10 μM ionomycin (right) for 10 min and subsequently incubated with Alexa-488-labeled Annexin V (green) to visualize exofacial PtdSer. Insets: pre-treatment images for each. **f** Comparison of the PtdSer content of plasma membranes isolated from control and mβCD-treated cells using polycationic beads. Plasma membranes were isolated as described in Supplementary Fig.1, and PtdSer quantified as in D. Data are means ± s.e.m of 4 separate experiments. **g** Subcellular distribution of PtdSer in control (black bars) and mβCD-extracted cells (open bars), determined by the colocalization using Pearson's correlation coefficient of the GFP-LactC2 probe with the following organellar markers: PH-PLCδ, (PM marker); Sec61, (endoplasmic reticulum marker); GalT, (Golgi complex); MitoTracker, (mitochondria); Rab5, Rab11 and LAMP1, used as markers of early endosomes, recycling endosomes and late endosomes/lysosomes, respectively. Data represent the means ± s. e.m of for 35 independent cells collected from three separate days. NS = not significantly different, *p < 0.05, **p < 0.01 and ***p < 0.005; as determined using a two-tailed t-test. The same test and designation of p-value is used throughout the remaining figures. Scale bar = 10 μm in all images

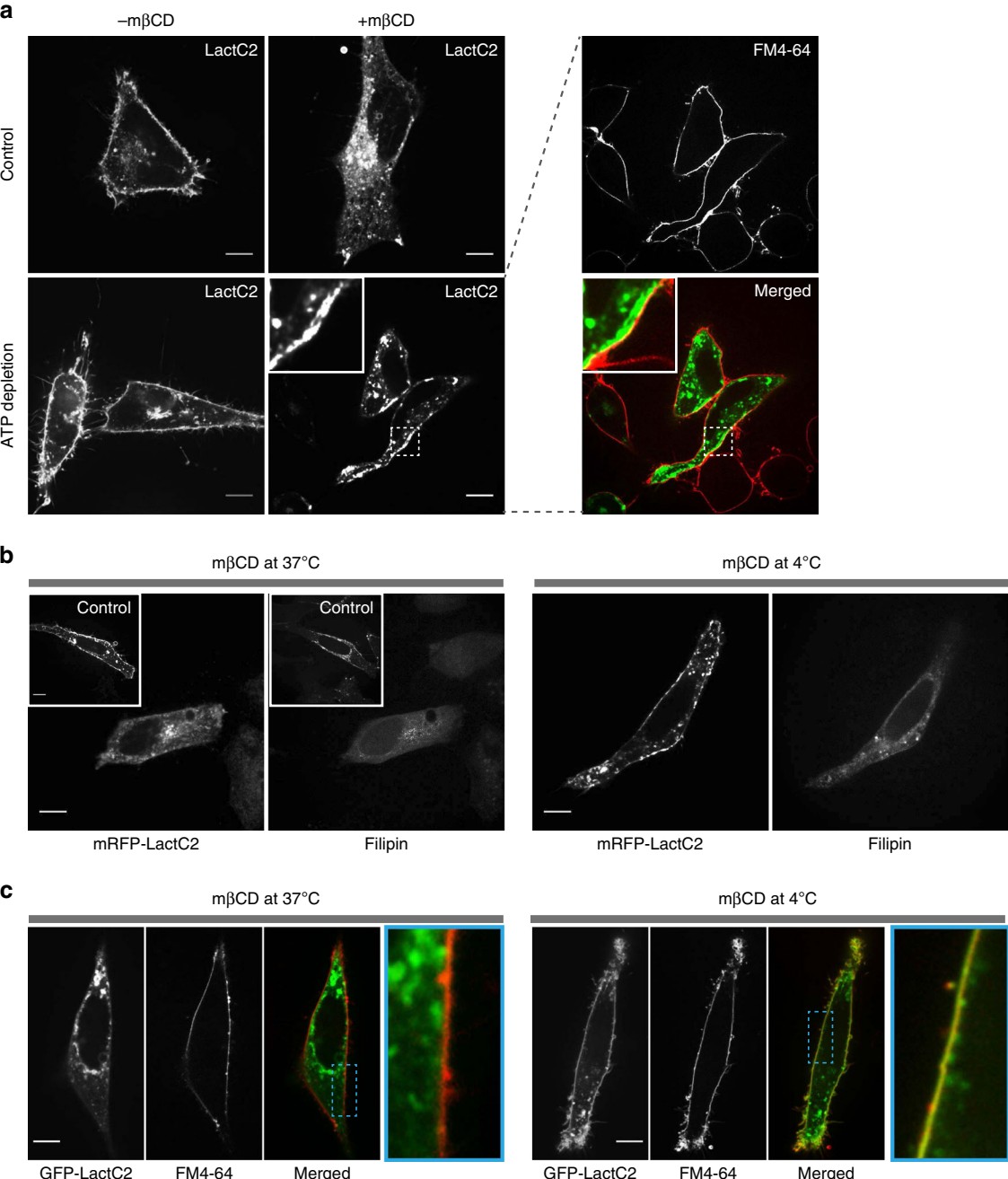

**Fig. 2** PtdSer internalization is ATP independent. **a** Representative images of HeLa cells expressing GFP-LactC2 treated with 10 mM mβCD for 30 min with or without ATP depletion, which was carried out pretreating cells with 0.2 μM antimycin A and 10 mM 2-deoxy-D-glucose for 60 min in $Ca^{2+}$-free and glucose-free medium. After mβCD treatment, cells were labeled with 40 ng/μL FM4-64. **b** Cholesterol extraction using mβCD was performed at 4 °C or 37 °C, as indicated, followed by fixation with 4% paraformaldehyde and labeling with 0.05% filipin. Inset: control cells without cholesterol removal. **c** mβCD was removed after 30 min and GFP-LactC2-expressing cells were labeled with FM4-64. Inset: magnifications of the area indicated by a dashed square. Scale bar = 10 μm in all images

represents ≈40 mol% of the plasmalemmal lipids, its removal is expected to decrease the surface area of the cell and increase the local concentration of the other lipids, including PtdSer. As PtdSer is anionic, increasing its concentration should increase the surface charge density of the cytosolic leaflet of the PM; this, in turn, could potentially facilitate endocytosis (see below). To investigate these concepts further, we switched to human red blood cells (RBCs), a simpler model system that lacks endomembranes and vesicular transport pathways. White (leaky) RBC ghosts make an attractive model as they retain an asymmetric PM, yet lose their hemoglobin due to the presence of small pores generated by hypotonic

treatment[13, 14]. Conveniently, these persistent pores also allow access of other small proteins and peptides to the ghost interior. Due to the loss of hemoglobin, ghosts are difficult to visualize using transmitted light but are readily visible using fluorescence microscopy when stained with FM4-64 (Fig. 3a). The surface area of nearly spherical ghosts was determined by acquiring serial optical sections and generating 3D projections (Methods section and Supplementary Fig. 3A–C). The surface area of the ghosts estimated by this approach averaged 132 μm$^2$, in good agreement with previous determinations[15, 16]. As mβCD rapidly extracts FM4-64, and the related FM1-43, from biological membranes

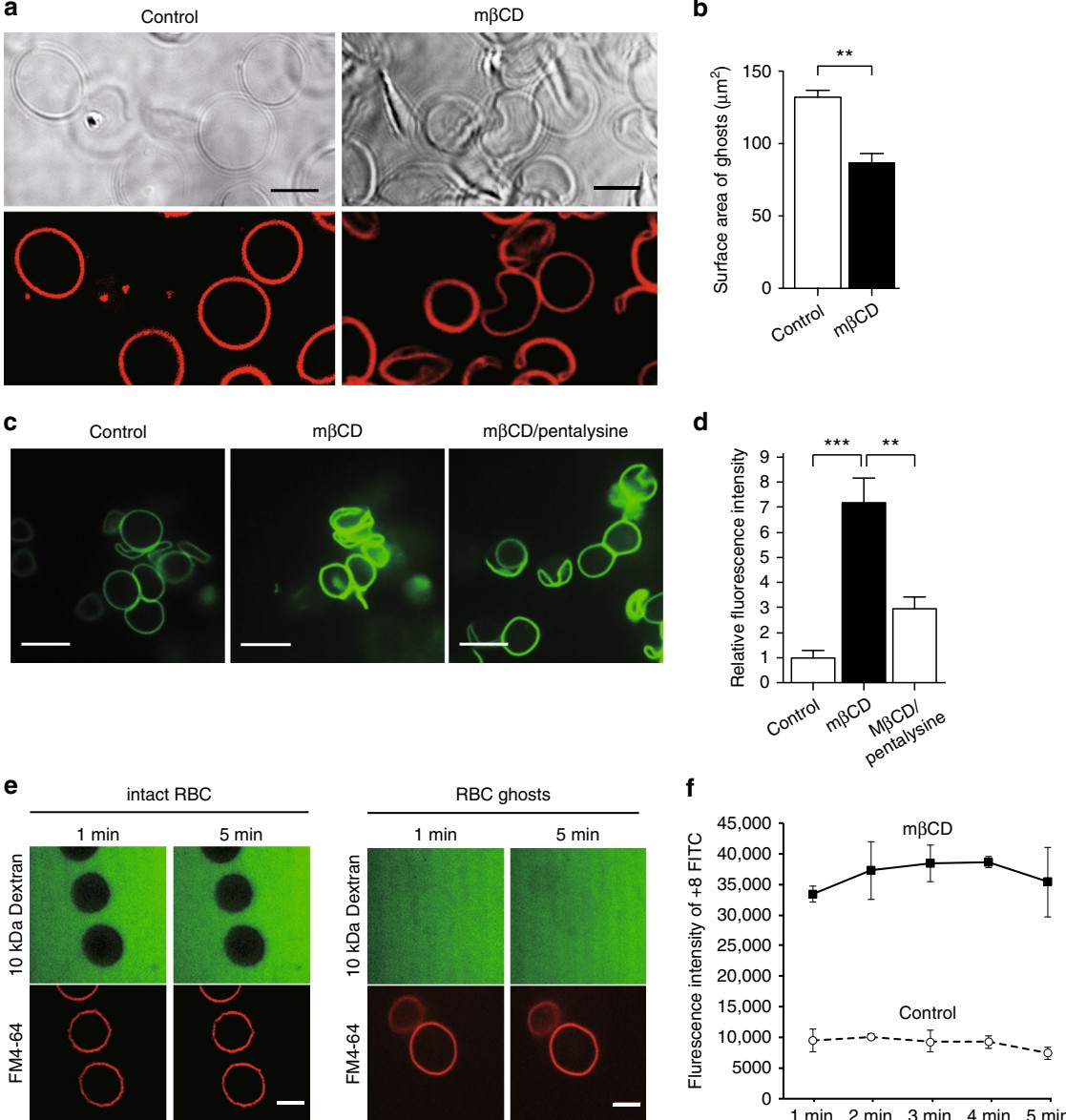

**Fig. 3** Cholesterol removal decreases surface area and increases the surface charge. **a** DIC (top) and fluorescence (bottom) images of RBC ghosts acquired by laser scanning microscopy. Ghosts before (left) and after (right) treatment with 10 mM mβCD for 1 min were labeled with 20 ng/μL FM4-64. Scale bar = 5 μm. **b** Surface area (in μm$^2$) of control and cholesterol-extracted ghosts. Data are means ± s.e.m., $n = 20$. **c** Representative images of FITC-tagged polycationic peptides electrostatically bound to the ghost cytosolic leaflet. Ghosts were bathed in 5P8 containing 6 ng/μL diluted peptides (left), and treated 10 mM mβCD for 1 min (middle), and with mβCD treatment after a preincubation with 1.5 mM pentalysine (right). **d** Relative fluorescence intensity of +8-FITC peptide bound to RBC ghost before or after mβCD treatment, with or without preincubation with pentalysine. The intensity of ghost membrane fluorescence was measured with ImageJ. Values represent the means ± s.e.m., $n = 41$. **e** Permeability of intact RBCs and ghosts to fluorescein conjugated 10 kDa dextran was assessed. Confocal images demonstrate the inability of dextran to enter intact RBCs and freely entering ghosts. **f** Time course of +8-FITC binding to the inner leaflet of control of cholesterol-depleted ghosts. The intensity of the membrane fluorescence was measured with ImageJ. Data represent the means ± s.e.m., $n = 3$. NS = not significantly different, *$p < 0.05$, **$p < 0.01$ and ***$p < 0.005$; as determined using a two-tailed t-test. Scale bar = 10 μm

(Supplementary Fig. 3D and refs. [17, 18]), the fluorescent probe was added to the ghosts after the cholesterol was extracted. After treatment with mβCD, the surface area declined to 87 μm$^2$, i.e. a ≈35% reduction in surface area (Fig. 3a, b).

A major determinant of the negative charge on the cytosolic surface of the PM is PtdSer, which constitutes ≈20% of the inner leaflet phospholipids. We hypothesized that reducing surface area would increase the lateral density of PtdSer (and other anionic phospholipids), thereby increasing the surface charge density. To monitor this change, we used a fluorescently-tagged surface

charge probe[19] consisting of an amphiphilic α-helix containing nine basic residues and a C-terminal carboxylic acid (+8-FITC). The exogenously added +8-FITC gains access to the cytosol of the ghosts and binds to the cytosolic leaflet of the membrane (Fig. 3c). It is noteworthy that +8-FITC does not bind to the exofacial leaflet of intact RBC (Supplementary Fig. 3D), due to the paucity of exofacial anionic lipids. As illustrated in Fig. 3c and quantified in 3D, + 8-FITC binding to mβCD-treated ghosts is considerably greater than to untreated ghosts. One possibility is that differences in membrane permeability could account for the

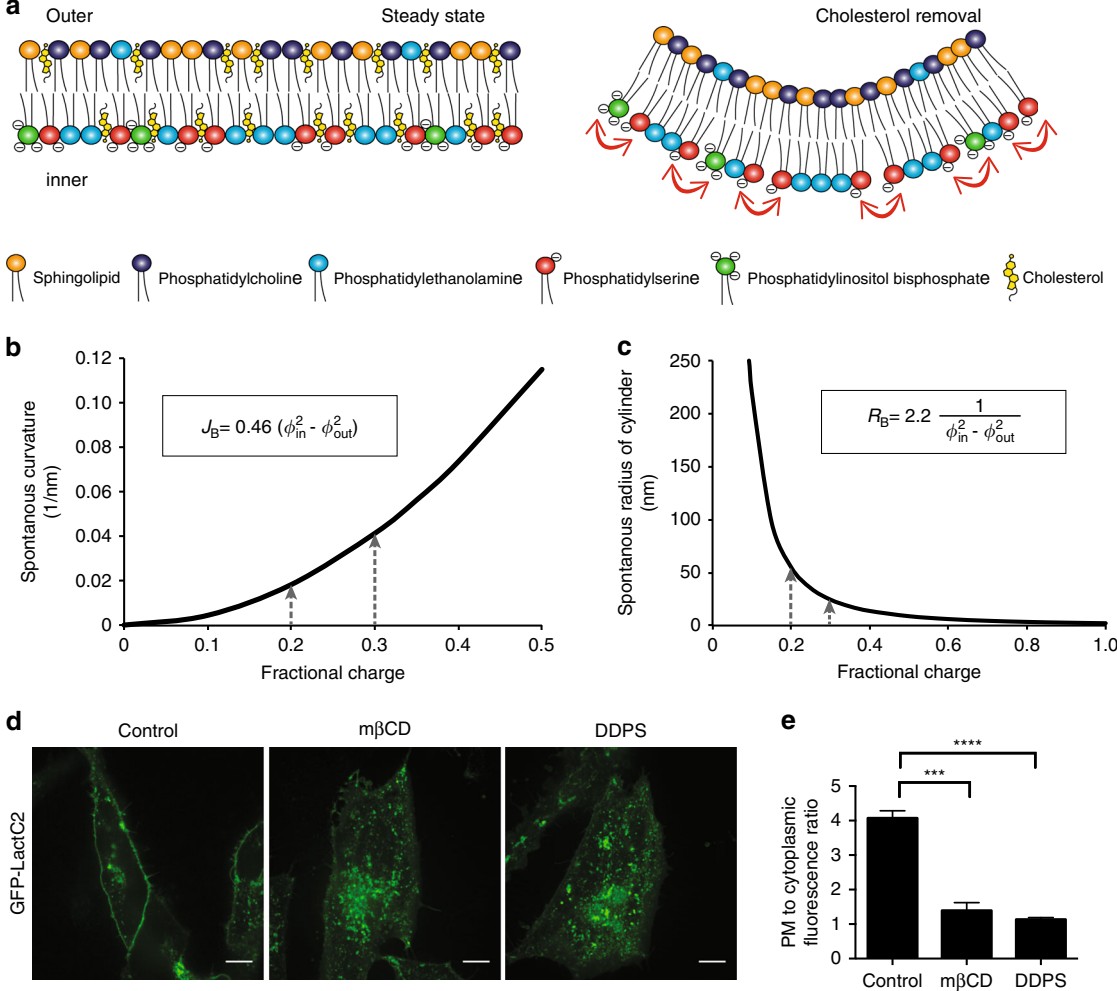

**Fig. 4** Increased charge density magnifies spontaneous curvature. **a** A diagram depicting a model membrane featuring phospholipids before (left) and after (right) cholesterol extraction. The color of the headgroup corresponds to each phospholipid, and the arrows indicate headgroup charge repulsion. **b** Spontaneous bilayer curvature predicted theoretically. $J_B$, a spontaneous curvature of a bilayer (nm); $\Phi_{in}$, fraction of charged lipids in the inner monolayer and $\Phi_{out}$, fraction of charged lipids in the outer monolayer. **c** Calculated spontaneous preferred cylinder radius ($R_B$) of a bilayer with varying fraction of charged lipids in the inner monolayer. **d** HeLa cells expressing the GFP-LactC2 probe were incubated with 10 mM mβCD or supplemented with 30 μM DDPS for 15 min and imaged using confocal microscopy. **e** Quantitation of the ratio of PM to cytoplasmic GFP-LactC2 in control cells, mβCD- or DDPS-treated cells. NS = not significantly different, *p < 0.05, **p < 0.01, ***p < 0.005 and ****p < 0.001; as determined using a two-tailed t-test. Values represent means ± s.e.m., n = 32

differential accumulation of the 2 kDa + 8-FITC peptide. However, this was not the case. We find that a significantly larger (10 kDa) dextran rapidly gains access to the cytosol of the control ghosts (Fig. 3e). In addition, in both control and cholesterol-extracted ghosts +8-FITC binding to the membrane reached a rapid equilibrium (Fig. 3f), indicating that the rate of permeation of the probe was not limiting the extent of its accumulation. That the excess binding to cholesterol-extracted ghosts is instead due to elevated surface charge is supported by the observation that pentalysine, a basic peptide that binds to and shields the charge of anionic lipids, markedly reduced the binding of the +8-FITC probe (Fig. 3c, d). Collectively, these findings support the notion that concentrating the anionic phospholipids by removal of cholesterol is associated with increased negative charge density, as predicted.

**Increased membrane charge density and spontaneous curvature.** The electrostatic repulsion of negatively charged lipid headgroups in the cytosolic leaflet can in principle induce spontaneous curvature in the PM (convex inward; Fig. 4a)[20, 21]. The relationship between spontaneous membrane curvature and charge density was estimated computationally as described in Supplementary Methods. Briefly, the lipid bilayer is considered as consisting of two monolayers, each with an overall surface charge derived from charged phospholipids. For the system parameter values used in Supplementary Method 1, the contribution to the spontaneous curvature of the bilayer related to the lipid charge is given by the expression

$$J_B = 0.46(\Phi_{in}^2 - \Phi_{out}^2)\,\text{nm}^{-1}$$

(Fig. 4b), where $\phi_{out}$ and $\phi_{in}$ denote the mole fractions of charged lipids within the outer and inner monolayer, respectively. The corresponding spontaneous radius of a bilayer cylinder is given by

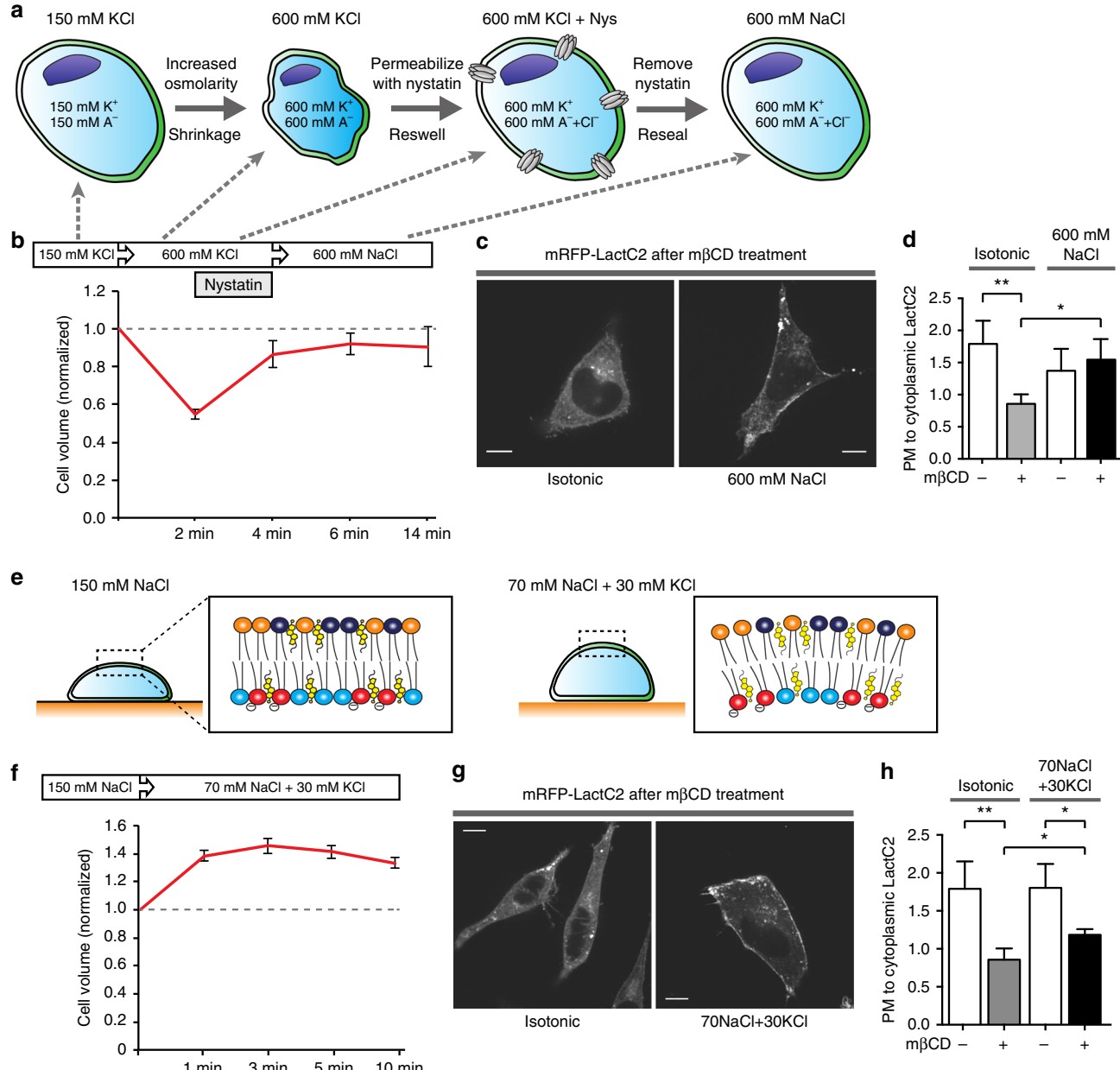

**Fig. 5** Increased ionic strength and membrane stretching prevent PtdSer internalization. **a** Neutralization of surface charge of the inner PM by increasing ionic strength. Increasing osmolarity by incubating in 600 mM KCl led to cell shrinkage. The addition of 40 ng/µL nystatin at low temperature allows entry of monovalent ions. To reseal the cells, nystatin is removed by washing with pre-warmed medium. Next cells were incubated in medium containing 600 mM NaCl to prevent further changes in cell size. To measure the cell volume, spinning-disk confocal imaging was performed on HeLa cells expressing mCherry. **b** The line graph depicts the changes in cell volume under the circumstances described in **a**. Values represent the means ± s.e.m. $n = 4$. All the media described in Fig. 5 contained 2 mM $CaCl_2$, 1 mM $MgCl_2$, 20 mM HEPES, 100 µM EGTA, 25 mM glucose, pH 7.4. [$A^-$] represents cellular anions. **c** HeLa cells expressing mRFP-LactC2 after treatment of 10 mM mβCD in an isotonic medium (left) and in 600 mM NaCl (right). **d** Quantitation of PM to cytoplasmic fluorescence ratio of mRFP-LactC2 after treatment of mβCD. **e** Sustained cell swelling in 70 mM NaCl + 30 mM KCl. Values represent means ± s.e.m. $n = 9$. **f** The line graph depicts the change of volume of cells incubated in medium containing 70 mM NaCl + 30 mM KCl. Values represent means ± s.e.m., $n = 4$. **g**, **h** Cells expressing mRFP-LactC2 after treatment of 10 mM mβCD in an isotonic medium (left) and in 70 mM NaCl + 30 mM KCl (right) for 10 min at 37 °C. NS = not significantly different, *p < 0.05, **p < 0.01 and ***p < 0.005; as determined using a two-tailed t-test. Data are means ± s.e.m., $n = 10$

the expression

$$\mathbf{R_B} = 2.2 \frac{1}{\phi_{in}^2 - \phi_{out}^2} \, \text{nm}$$

(Fig. 4c). As a result of cholesterol extraction, the mole fraction of charged lipids in the inner monolayer, $\phi_{in}$, is estimated to rise

from 0.2 to 0.3, while the amount of charged lipids in the outer monolayer is so small, that its effects can be neglected, $\phi_{out} = 0$. Assuming that the surface charge density of the inner monolayer is the only reason for the bilayer spontaneous curvature, the latter would change from about 0.018/nm, which is nearly flat, to 0.041/nm, representing a significant degree of bending. The radius of a cylinder arising from this spontaneous curvature would decline

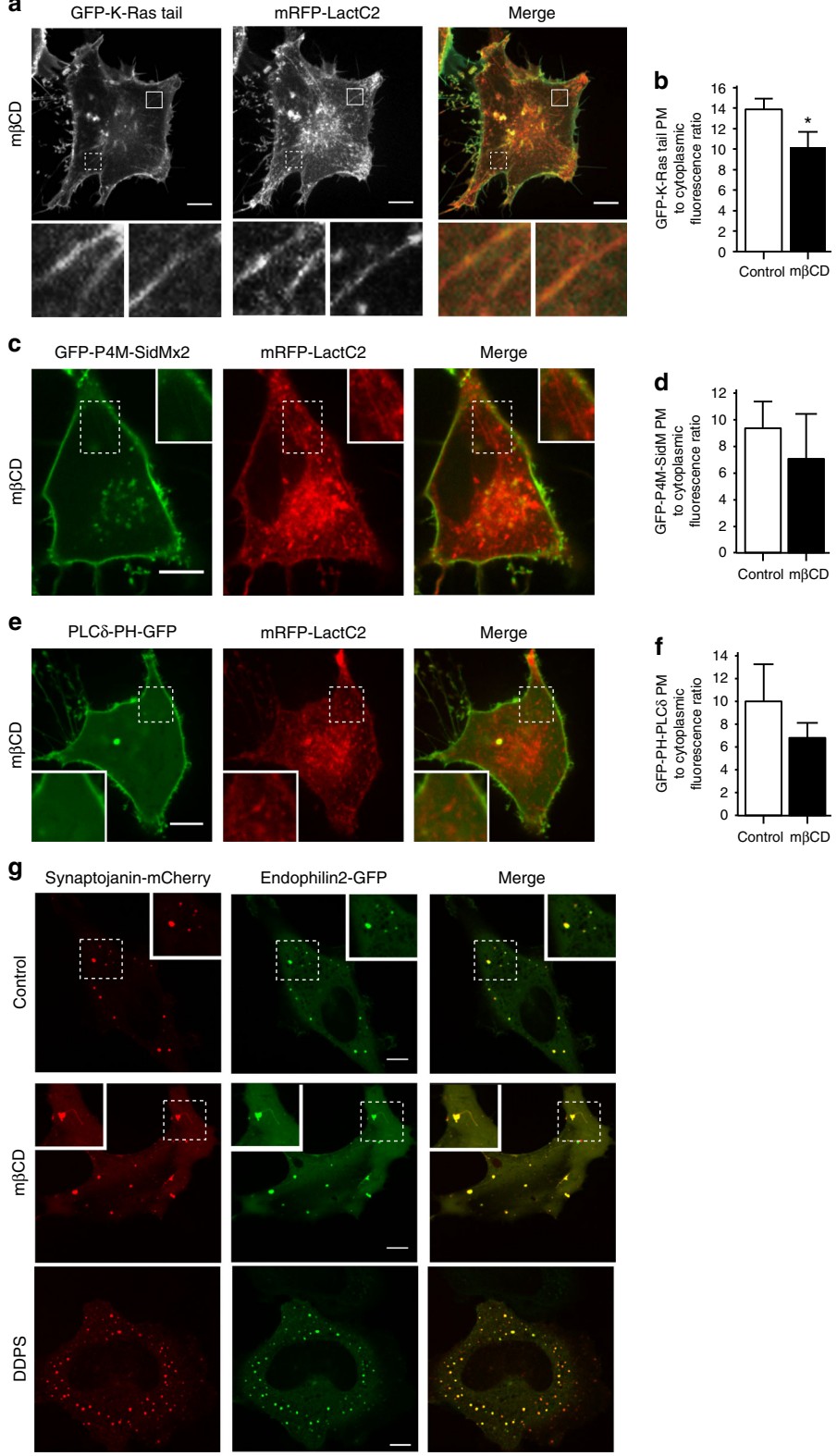

**Fig. 6** Acute removal of cholesterol forms PtdSer-rich tubules. **a** Confocal images of following mβCD treatment of HeLa cells co-transfected with mRFP-LactC2 and GFP-K-Ras tail. Lower left insets: magnification of the areas indicated by a dashed square. Lower right insets: magnification of areas indicated by a lined square. **b** Quantitation of the distribution of GFP-K-Ras tail between the plasma membrane and the cytoplasm 20 min after the addition of mβCD. Data are means ± s.e.m., $n = 14$. **c**, **e** mβCD treatment of HeLa cells co-transfected with mRFP-LactC2 and either the PtdIns4P probe GFP-P4M-SidMx2 or the PtdIns4,5P$_2$ probe GFP-PLCδ-PH. Insets: magnification of the areas indicated by a square. **d**, **f** Quantitation of the distribution of GFP-P4M-SidMx2 or GFP-PLCδ-PH between the plasma membrane and the cytoplasm 20 min after the addition of mβCD. Values represent the means ± s.e.m., $n = 11$.
**g** Confocal images of control, DDPS and acute mβCD-treated HeLa cells co-expressing Synaptojanin2-mCherry and Endophilin2-GFP. NS = not significantly different, *$p < 0.05$, **$p < 0.01$ and ***$p < 0.005$; as determined using a two-tailed t-test. Scale bar = 10 μm

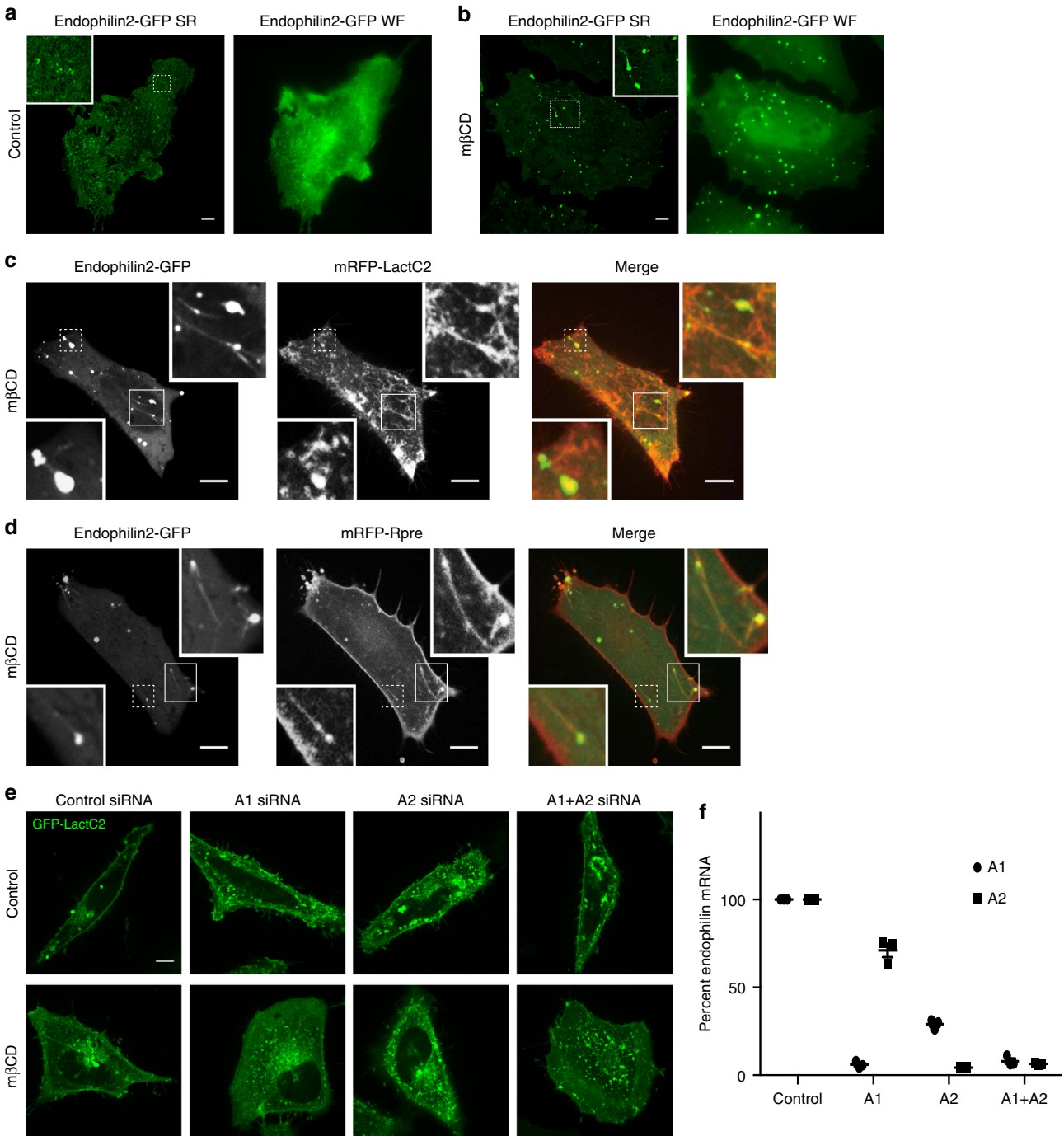

**Fig. 7** Endophilin is recruited to sights of internalization but is not required. Control (**a**) and mβCD-treated (**b**) HeLa cells expressing endophilin2-GFP were images using wide-field (WF) and super-resolution structured illumination microscopy (SR). **c**, **d** Confocal images of HeLa cells co-expressing Endophilin2-GFP and either mRFP-LactC2 or the charge probe mRFP-Rpre following the addition of mβCD for 2 min. Insets: magnification of the areas indicated by a square. **e** HeLa cells were co-transfected with the indicated Stealth RNA (control, endophilin A1, endophilin A2, or both) and GFP-LactC2. Confocal images of control cells and cells 20 min following the addition of mβCD. **f** Quantitation of Endophilin A1 and A2 mRNA transfected following transfection with control and endophilin A1 and A2 Stealth RNA. Data represent the means ± s.e.m., $n = 3$

from about 55 nm to 24 nm. Importantly, this degree of curvature can be captured, stabilized, and/or enhanced by curvature-sensing/binding protein domains, e.g., BAR domains[22]. On the basis of these considerations, we hypothesized that cholesterol removal results in an increase in charge density, leading to enhanced headgroup repulsion and an intensification of spontaneous membrane curvature. This, in turn, may foster tubulation of the PM.

**Addition of PtdSer is sufficient to stimulate endocytosis**. The model predicts that a sufficient increase in anionic charge density in the inner leaflet of the PM should promote the formation of spontaneous curvature and support the initiation of endocytosis. To test this prediction in a manner independent of cholesterol extraction, we supplemented cells with PtdSer. Previous results have demonstrated that exogenously added PtdSer is effectively incorporated into the exofacial leaflet of the PM and rapidly

flipped to its cytosolic leaflet[23, 24]. To monitor the distribution of PtdSer following addition of the lipid we used cells transfected with the GFP-LactC2 probe. As depicted in Fig. 4d and quantified in Fig. 4e, in HeLa cells incubated with 30 μM 1,2-didecanoyl-PtdSer (DDPS) for 15 min the LactC2 probe underwent relocalization to endomembranes, recapitulating the observations made with mβCD. These results demonstrate that an increase in cytosolic leaflet PtdSer, and thus anionic charge, can stimulate internalization of the plasma membrane, regardless of whether the change was induced by insertion and flipping of exogenous PtdSer, or by cholesterol removal.

**Ions mitigate headgroup repulsion and PtdSer internalization.** The magnitude of charge repulsion will be dictated in part by the ionic concentration of the surrounding medium (i.e., the cytosol). Thus, we sought to determine if increasing the cytosolic ionic strength—by changing the intracellular ionic concentration—would prevent the internalization of PtdSer. To this end, we used nystatin, a reversible pore-forming ionophore that enabled us to alter the ionic content of the cells with negligible changes in cell volume despite quadrupling the concentration of monovalent ions to 600 mM (see Fig. 5a, b and legend for details of the protocol and measurements of cell volume). When the ionic strength was elevated in this manner the mβCD-induced redistribution of PtdSer was abolished (Fig. 5c, d). Much like the depletion of cellular ATP, this maneuver to increase the ionic strength of the cytosol may influence the functionality of some proteins. However, the finding is consistent with the notion that increased ionic strength can shield the negatively charged headgroups of the phospholipids, limiting the charge repulsion and the induction of curvature that leads to endocytosis.

**Plasma membrane tension attenuates PtdSer internalization.** If PtdSer removal from the plasma membrane is indeed mediated by the membrane budding and endocytosis, this process should be antagonized by distending the membrane[25]. We tested this premise by swelling the cells osmotically; a steady degree of swelling was imposed by lowering the osmolarity while keeping the [K][Cl] product constant to preclude regulatory volume decrease[26]. We demonstrated that bathing the cells in medium containing 70 mM NaCl and 30 mM KCl caused the cells to swell by ≈35% and that this swelling was sustained for at least 10 min (Fig. 5e, f). Remarkably, when the membrane was stretched, the retention of LactC2 at the membrane following cholesterol extraction was significantly better than in extracted but otherwise untreated (unstretched) cells (Fig. 5g, h).

**PtdSer-rich tubules emanate from the PM upon sterol removal.** To better understand the mechanism responsible for PtdSer loss from the membrane, we monitored the fate of LactC2 continuously during cholesterol depletion. Within 1–2 min of treatment with mβCD, we noted the formation of PtdSer-rich tubules emanating from the plasma membrane (Fig. 6a). These tubules were highly negatively charged, as indicated by the distribution of the C-terminal tail of K-Ras (K-Ras tail), a polycationic ( + 8) surface charge probe that identifies anionic membranes within the cell (Fig. 6a)[11, 19]. When quantified 15–20 min after the addition of mβCD, after the tubules have largely disappeared, not only did PtdSer relocalize to endomembranes, but so did a fraction of the surface charge indicator (Fig. 6b). Remarkably, other negatively charged lipids, such as PtdIns(4)P or PtdIns(4,5)P$_2$ were not detectable in the tubules or foci generated by extraction of cholesterol; this was established using the P4M-SidMx2 and PH-PLCδ fluorescent biosensors, which monitor freely accessible ligands on the cytosolic leaflet of cellular membranes (Fig. 6c, e

and Supplementary Fig. 4). This suggests that the high negative surface charge of the tubules is attributable to the accumulation of PtdSer. The concomitant loss of negative charge from the membrane is detectable when using a surface charge detector with +6 charges (Supplementary Fig. 5A), but is less obvious with the +8 probe that is more sensitive to the elevated charge density of the phosphoinositides[11, 19], which were not significantly depleted from the plasma membrane following the loss of plasmalemmal cholesterol (Fig. 6d, f). Collectively, our data suggest that the electrostatic repulsion of charged lipids can account for the generation of small areas of inward (convex) curvature that can extend to generate cylindrical tubules.

The absence of PtdIns(4)P and PtdIns(4,5)P$_2$ from the tubules was an unexpected, yet reproducible observation. We speculated that synaptojanin, a dual inositol 5-phosphatase and 4-phosphatase that regulates membrane fission during endocytosis could be responsible for the disappearance of the phosphoinositides. Importantly, the activity of synaptojanin is stimulated on highly curved membranes, in part due to interactions with endophilin, an N-BAR (Bin/Amphiphysin/Rvs) domain-containing protein[27, 28]. Of note, BAR domains associate preferentially with curved membranes and, in some instances, interact directly with anionic phospholipids, including PtdSer[29]. The conditions generated by extraction of cholesterol are, in principle, conducive to the recruitment of endophilin and activation of synaptojanin. We analyzed the localization of synaptojanin2 and endophilin2 in cells treated with mβCD. As shown in Fig. 6g, while some synaptojanin and endophilin are normally associated with resting membranes in HeLa cells, cholesterol extraction-induced a marked increase in the size of the foci containing these proteins. Similarly, the addition of DDPS also increased the number of foci marked by both synaptojanin2 and endophilin2. Strikingly, tubular structures were seen to emerge from the synaptojanin2- and endophilin2-enriched foci (Fig. 6g). Thus, the recruitment of synaptojanin can account, at least in part, for the lack of PtdIns(4)P and PtdIns(4,5)P$_2$ in the tubules.

The formation of large foci of endophilin would seem at odds with our theoretical calculations that predict the formation of tubules with a radius of 25–30 nm upon cholesterol removal (see above). It is possible, however, that such tubules undergo coiling near the membrane, giving the appearance of large foci when analyzed by conventional (diffraction-limited) microscopy. We used structured illumination microscopy (SIM) to better resolve the endophilin-positive structures (Fig. 7a and Supplementary Movie 1). Unlike the small endophilin puncta seen in untreated cells, the larger structures formed following cholesterol extraction had irregular shapes, often appearing as multi-lobed structures with tubules emerging from some of them (Fig. 7b and Supplementary Movies 1 and 2). These observations are entirely consistent with the serpentine clusters of tubules observed previously by electron microscopy in cholesterol-depleted cells[30].

Because endophilin was shown to mediate clathrin-independent endocytosis via tubulovesicular carriers[31, 32], we considered whether it was similarly involved in facilitating the tubulation and endocytosis that accompanies cholesterol extraction and PtdSer compaction. This possibility was given credence by the finding that endophilin2-GFP demarcates the tubules identified by the PtdSer probe and the negative charge detector mRFP-RPre (Fig. 7c, d). To test the causal relationship between these events we silenced the endophilin A1 and A2 genes. Using Stealth RNAi technology (Invitrogen) we effectively reduced the expression of both endophilin A1 and A2 (Fig. 7f). Despite the nearly complete silencing of the two endophilin isoforms, treatment with mβCD nevertheless produced internalization of plasmalemmal PtdSer. Thus, while endophilin is a marker of the

tubules generated by cholesterol depletion, it is not essential for either the tubulation or endocytosis processes.

**Verotoxin-induced endocytosis does not require PtdSer.** Verotoxin—an AB-type bacterial toxin—binds the exofacial lipid, globotriaosylceramide (Gb3), inducing invaginations of the PM that lead to endocytosis[33–35]. It is unclear whether PtdSer or changes in anionic charge density of the inner leaflet are required to support this mode of internalization. As shown in Supplementary Fig. 6A, the addition of sphingosine, a positively charged lipid, completely inhibited the internalization of verotoxin, without preventing its binding to Gb3. This is consistent with a role of negative charge in the uptake mechanism. Additionally, we noted that tubules induced by verotoxin are rich in PtdSer, which is compatible with the notion that they are negatively charged (Supplementary Fig. 6B).

To more directly address the requirement for PtdSer for verotoxin-mediated endocytosis, we assessed the effect of varying its concentration on toxin internalization. While complete elimination of PtdSer (e.g. by deletion of both PtdSer synthases 1 and 2) is lethal, cells with considerably reduced synthase activity are viable. We therefore used a Chinese hamster ovary (CHO) cell line, termed PSB-2, that has a $\approx 95\%$ reduction in PtdSer synthase activity, which translates to a $\approx 75\%$ reduction in PtdSer levels. CHO (and hence also the PSB-2) cells lack Gb3, the ligand recognized by verotoxin. To circumvent this limitation we transfected α1,4-glycosyltransferase (A4GALT; heterologous expression of this gene was shown earlier to be sufficient to catalyze Gb3 production in CHO cells[36]. Introducing the A4GALT into wild-type and PSB-2 cells allowed us to examine whether PtdSer plays a role in verotoxin uptake. As shown in Supplementary Fig. 6C, D, verotoxin-induced endocytosis persisted in PSB-2 cells despite their markedly reduced PtdSer content. Thus, while charge neutralization using sphingosine inhibited uptake, the reduction in PtdSer had minimal impact. However, it is worth noting that the depletion of PtdSer in PSB-2 cells is accompanied by a compensatory increase in the levels of phosphoinositides[37], which are anionic. Thus, while PtdSer per se is not necessary for optimal internalization of verotoxin, a negative inner surface charge may nevertheless be required to support the process.

## Discussion

Our data indicate that sudden removal of cholesterol triggered tubulation and endocytosis of PtdSer-enriched regions of the plasma membrane. The unique shape of cholesterol can itself influence the curvature of the membrane[26]. However, in principle, this effect would be exerted equally on both leaflets of the bilayer, with offsetting effects. Although the distribution of cholesterol across the plasmalemma remains the subject of debate[7, 38], our data suggest that rather than contributing asymmetrically to membrane distortion because of its peculiar shape, cholesterol influences membrane curvature by modulating the spacing between anionic phospholipids. We propose that the appearance of curved domains upon treatment with mβCD arises from the highly asymmetric distribution of phospholipids across the plasmalemmal bilayer, where the anionic phosphoinositides and PtdSer are restricted almost exclusively to the inner leaflet. We believe that inward curvature develops upon cholesterol extraction as a result of electrostatic repulsion between the headgroups of anionic phospholipids, which are brought closer together when the spacing normally provided by cholesterol is eliminated. Three lines of evidence support this contention: first, estimation of the charge density of the inner surface of red cell ghosts using fluorescent sensors showed a marked increase when cholesterol

was extracted (Fig. 3c, d). Accordingly, in HeLa cells we detected increased recruitment of endophilin—a protein reported to associate electrostatically with anionic membranes—to the PM following cholesterol extraction. Secondly, the internalization of PtdSer was greatly attenuated when the ionic strength of the cytosol was elevated, shielding the charges of the phospholipid headgroups (Fig. 5c, d). Finally, the addition of exogenous PtdSer to otherwise untreated cells is also a potent stimulator of PtdSer endocytosis.

Our calculations (Fig. 4b, c) reveal a surprisingly steep relationship between the curvature of the "dimples" that are expected to form spontaneously in the membrane as a consequence of electrostatic repulsion, and the surface charge density expressed as the mole fraction of charged lipids on the membrane surface. The predicted radius of the inward tubules decreases sharply as the fractional charge increases, from $\approx 55$ nm at a charge density that we believe approximates that of the inner leaflet of the plasma membrane of resting cells (based on the reported lipid composition), to near $\approx 24$ nm when the fractional charge is increased by 50%. Strikingly, the increased curvature generated by altering the charge density can be captured and stabilized by BAR domains like that of endophilin, which also would be attracted and retained electrostatically. The removal of cholesterol would, by itself, add a measure of stability to the induced curvature. Bruckner and colleagues[39] suggested that, by virtue of its ability to spontaneously flip-flop rapidly between leaflets of the bilayer, cholesterol can reduce the bending energy of the membrane. The stress-relaxing effect normally provided by the directional flip-flop that is induced when curvature is imposed[39], would be absent following extraction with mβCD. Thus, the rapid extraction of cholesterol could foster tubulation and endocytosis in a variety of ways.

Recruitment of endophilin and synaptojanin to the membrane were initially observed during the course of clathrin-mediated endocytosis. Paradoxically, removal of cholesterol was reported to inhibit transferrin receptor internalization and the appearance of clathrin lattices[40, 41]. While the underlying mechanism is not understood, inhibition of clathrin-mediated endocytosis does not conflict with the increased tubulation and endocytosis reported here. First, while dynamin is essential for the scission of clathrin-coated pits from the PM, inhibition of dynamin with Dyngo4A did not prevent the internalization of PtdSer following cholesterol extraction (Supplementary Fig. 7). Moreover, the internalization of PtdSer induced by mβCD precedes the impairment of clathrin-dependent endocytosis, which was measured 15–30 min after cholesterol extraction. It is also noteworthy that, while initially associated with clathrin-mediated endocytosis, endophilin has more recently been implicated in other modes of membrane internalization, including a fast clathrin-independent endocytosis[31], and in the uptake of Shiga and cholera toxins[32]. In all instances, endocytosis is triggered by the development of membrane curvature that is stabilized by association with the crescent-shaped N-BAR domain of endophilin, as suggested to occur upon cholesterol depletion. However, at least in the latter case, while the association of endophilin with the membrane is indicative of developing curvature, it is not essential for the formation of tubules of for their scission from the PM (Fig. 7).

The extensive removal of cholesterol obtained when treating cells with mβCD and the accompanying global elevation in charge density are clearly unphysiological. However, cholesterol removal served to illustrate the potential effects of increased charge density on membrane curvature. Physiologically, focal accumulation of negative charge can be envisaged to occur under a variety of circumstances. Negativity can increase locally by activation of lipid (e.g., phosphoinositide or sphingosine) kinases or hydrolases (e.g., phospholipase D), by segregation of domains induced by

protein or lipid redistribution (e.g., upon clustering of immunoreceptors), and/or by localized flipping of anionic lipids (e.g., at sites where exocytosis of vesicles containing luminal PtdSer occurs)[30, 42, 43]. Our results also further elucidate the relationship between cholesterol and PtdSer. The relative abundance of these two lipids among cellular organelles is remarkably parallel, but the explanation for this coincident behavior is not well understood. While cholesterol is often thought to interact with sphingomyelin in the exofacial leaflet of the PM, underlying the establishment of liquid-ordered nanodomains, the rationale for its presence and the forces retaining it in the inner leaflet are less clear. Our results suggest that cholesterol is interspersed between PtdSer molecules, which in the PM are generally saturated, likely forming ordered nanodomains[8, 9]. The large headgroup of Ptdser can shield the more hydrophobic cholesterol from water while the spacing afforded by cholesterol separates the charged PtdSer molecules. Moreover, the spontaneous negative curvature introduced by the shape of cholesterol counteracts the positive curvature imposed by electrostatic repulsion of anionic phospholipids. PtdSer-enriched nanodomains are indeed present at the membrane, as shown recently by electron microscopy[5]. Thus, the maintenance of high (physiological) levels of PtdSer in the inner leaflet seemingly requires this association with cholesterol to prevent the coalescence of the anionic headgroups and the development of excessive repulsion, curvature, and internalization.

In summary, our study highlights electrostatic repulsion as a source of membrane curvature leading to endocytosis, and provides a rationale for the co-distribution of PtdSer and cholesterol in cellular membranes.

## Methods

**Reagents**. Alexa-488-conjugated Annexin-V and MitoTracker Green were purchased from Thermo Fisher Scientific (Waltham, MA). D-erythro-sphingosine was from Cayman Chemical Company (Ann Arbor, MI). FM4-64 was purchased from Life Technologies (Carlsbad, CA). All other fine chemicals were from Sigma-Aldrich (St. Louis, MO). Recombinant Cy3-labeled verotoxin B subunit was provided by Dr. C. Lingwood (Hospital for Sick Children, Toronto, Canada). Short-chain PtdSer, 1,2-didecanoyl-sn-glycero-3-[phospho-L-serine] (DDPS 10:0) was purchased from Avanti Polar Lipids (Alabaster, USA).

**Cell culture and transfection**. HeLa cells obtained from American Type Culture Collection (Manassass, VA) were incubated in Dulbecco's modified Eagle medium (DMEM) with 5% fetal bovine serum (FBS, from Wisent) at 37 °C under 5% $CO_2$. Transient transfection was performed using FuGene 6 (Promega) according to the manufacturer's instructions. Briefly, each well of a 12-well plate was treated with 1 µg plasmid cDNA and 1.5 µL FuGene 6. Transfected cells were used 18–24 h after addition of transfection mixture. For live cell imaging, cells were incubated in HEPES-containing HBSS or a medium containing 140 mM NaCl, 5 mM KCl, 2 mM CaCl₂, 1 mM MgCl₂, 20 mM HEPES, 100 µM EGTA, 25 mM glucose, pH 7.4 at 37 °C. Cell are routinely tested for mycoplasma contamination using e-Myco Real-Time PCR Detection Kit (FroggaBio, Toronto, Canada).

For the addition of exogenous PtdSer to cells, DDPS was prepared by drying the lipid under a stream of nitrogen and resuspended in PBS. Subsequently, the solution was sonicated in a water bath for 15 min. HeLa cells were treated with 30 µM DDPS in serum-free medium for 15 min, images were acquired in live cells.

**Plasmids**. Construction of the plasmids encoding Lact-C2[11], PH-PLCδ[44], P4M-SidMx2[45], R-pre[19], K-Ras tail[46], Sec61[47], GalT[48], Rab5[48], Rab11[49] and LAMP[50] have been described previously. Human pCMV-SPORT6-A4GALT (Gb3 synthase) and endophillin-2 were obtained from the Harvard plasmid repository. The endophilin-2 plasmid was used as a template for PCR using the following pairs of primer: 5′-GCGAGATCTATGTCGGTGGCGGGGCTAAA-3′ (endophillin-2 sense primer) and 5′-CGCGGATCCTTCTGCGGGCAGGGGCACAAGCA-3′ (endophillin-2 antisense primer). The PCR product was introduced into the multiple cloning site in the pEGFP-N1 vector using the restriction enzymes BglII and BamHI.

**siRNA-mediated silencing and qRT–PCR**. Stealth RNAi (siRNA) directed against SH3GL1 and SH3GL2 and previously validated[31] were purchased from Invitrogen. For siRNA transfection of HeLa cells, 12 pmol of siRNA and 1.6 µL of Lipofectamine RNAiMAX transfection reagent (Invitrogen) was used according to the

manufacturer's protocol. Knockdown efficiency was determined 48 h post transfection.

To quantify the expression of endophilin mRNA after siRNA knockdown, RNA was purified from HeLa cells using the GeneJet RNA purification kit (Thermo Fisher Scientific). This was used as a template for cDNA synthesis using the Superscript VILO cDNA synthesis kit (Invitrogen). The endophilin A1-specific and endophilin A2-specific Taqman gene expression assay (Life Technologies) was used for real-time quantitative PCR and it was normalized relative to CDKN1 mRNA.

**Lipid determination**. Lipids were extracted by the Folch method[51] and reacted with fluorescamine[52]. The isolated lipids, along with known amounts of pure lipids (Avanti Polar Lipids) were separated by thin layer chromatography on 250 µm silica gel TLC plates (Whatman, Maidstone, UK) using a resolving solution of CHCl₃, MeOH, CH₃COOH and H₂O (25:15:4:1). The fluorescamine-labeled lipids were visualized with a Storm 840 chemiluminescence imager system (Molecular Dynamics). Fluorescence intensities were used to calculate the amount of isolated lipid from a standard curve constructed with the lipid standards. The amount of isolated lipid was then normalized to the quantity of protein from the entire cell, or only the PM, from which the lipids were isolated.

**LactC2 liposome-binding measurements by FRET**. Binding of GST-LactC2 to liposomes was measured via FRET from intrinsic tryptophan residues to dansyl-labeled PE, as described previously[53]. Briefly, experiments were performed in a 5 × 5 × 30 mm quartz cuvette in an F-2500 Fluorescence Spectrophotometer (Hitachi, Tokyo, Japan) at room temperature. Excitation and emission wavelengths were 280 and 510 nm, with 5 nm slit widths. The total lipid concentration for the measurement was adjusted to 10 µM, GST-LactC2 was added in increments as depicted on the graph, and the fluorescence recorded and corrected for dilution. Data were analyzed as previously described[53] using the equation $F_c = F \div F_b - 1$ where $F_b$ is dansyl fluorescence in the absence of LactC2, F is the fluorescence at a given protein concentration and $F_c$ is the corrected fluorescence (i.e., background subtracted) at a given protein concentration.

**Preparation of RBC and unsealed ghosts**. Fresh venous blood drawn from healthy volunteers was treated with 50 U/mL heparin, washed with PBS twice and resuspended in five volumes of PBS. Hemolysis was initiated by thoroughly mixing 1 mL of the prepared red blood cells with 40 mL of 5P8 buffer (5 mM sodium phosphate, pH 8.0). The membranous ghosts were collected by centrifugation at $25,000 \times g$ for 10 min in a fixed-angle rotor. The resulting supernatant containing free hemoglobin was removed. Tubes were tilted and rotated to allow the loose ghost pellets to slide off the wall and then ghosts were carefully collected. Ghost membranes were subjected to additional wash cycles with 5P8 buffer until white[13, 14]. All procedures were performed at 4 °C. Coverslips were treated with 0.1% concanavalin A and washed twice with 5P8 buffer to immobilize the ghost membranes for visualization.

**Measurement of surface area in RBC ghosts**. Confocal images of ghosts were acquired at multiple planes by laser scanning microscopy and the surface area (µm²) at the base was measured using ImageJ (National Institutes of Health, Bethesda, MD). The rest of the area was calculated as the sum of the surface areas of the cylindrical optical sections, which were determined as the product of the height (0.4 µm, the thickness of the optical slices) times their circumference (Supplementary Fig. 3).

**Peptide synthesis**. The polycationic +8 peptide was custom-synthesized by Biosynthesis, Inc. (Texas, USA). The amino acid sequence, basic residues in red (SKLKRLFKRLRKWFKKGC-COOH) was labeled with FITC at the N-terminal end and purified to ≥95% purity. The RBC ghosts were incubated with the peptide at 6 ng/µL in 5P8 buffer.

**Microscopy**. Fluorescence images were acquired by spinning-disk confocal microscopy (Quorum Technologies). An Axiovert 200M microscope (Carl Zeiss) with 63 × objective lenses and a × 1.5 magnifying lens, equipped with diode-pumped solid-state lasers (440, 491, 561, 638, and 655 nm; Spectral Applied Research) and a motorized XY stage (Applied Scientific Instrumentation). Images were acquired using back-thinned, electron-multiplied cameras (model C9100-13 ImagEM; Hamamatsu Photonics) driven by the Volocity software (PerkinElmer). DIC images of ghosts were acquired using a 100×/1.4 oil immersion objective on a Nikon A1R Si confocal microscope. Fluorescent images of ghosts labeled by FM4-64 were taken using 561 nm laser excitation.

For Structured Illumination Microscopy (SIM) images were acquired using the Zeiss Elyra PS1 system with an Axio Observer Z1 microscope, 63× (1.4 NA) oil immersion objective, 1.6× optovar and an Andor iXon3 885 EM-CCD camera, using the ZEN software. The microscope is equipped with a Zeiss motorized XY stage and Z-piezo focus drive. Three orientation angles of the excitation grid with five phases each were acquired for each x plane. Images were reconstructed using the structured illumination module in the ZEN software.

**Statistics**. For statistical analysis, paired or unpaired t-tests were used with a 95% confidence limit consistent with the practice in this field. All data presented in the text and graphs are means ± s.e.m with the number of samples indicated. Samples were acquired over a minimum of three separate days.

**Data availability**. The data that support the findings of this study are available from the corresponding author upon reasonable request.

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

## Acknowledgements

This work was supported by an Operating Grants MOP-133656 and a Foundation Grant FDN-143202 from the Canadian Institutes of Health Research (CIHR) to G.D.F and S.G., respectively. G.D.F. is also a recipient of a New Investigator Award from the CIHR and an Early Researcher Award from the Government of Ontario. The authors declare no support from any organization for the submitted work; no financial relationships with any organizations that might have an interest in the submitted work; no other relationships or activities that could appear to have influenced the submitted work.

## Author contributions

T.H. designed and performed the experiments, analyzed the data and assembled figures. S.M.L. designed and performed experiments, analyzed the data and generated the animations. M.M. performed experiments and analyzed data. M.M.K. performed the biophysical calculations regarding spontaneous curvature and wrote the manuscript. J.G.K., S.G. and G.D.F. conceived the project, designed the experiments, analyzed the data and wrote the manuscript.

## Additional information

**Competing interests:** The authors declare no competing financial interests.

