## [Peer Review File · Nature Communications]

Reviewers' Comments:

Reviewer #1 (Remarks to the Author)

This manuscript presents a novel hypothesis for the simultaneous presence of the anionic lipids phosphatidylserine and cholesterol in the inner leaflet of the plasma membrane: cholesterol would attenuate the lateral repulsion between PS molecules, which in the absence of phosphatidylserine have a tendency to curve the plasma membrane inward. This effect can be advantageous for endocytosis. The manuscript includes a physical model to estimate the effect of cholesterol removal on PS induced membrane curvature and various cellular assays. They combine cyclodextrin to remove cholesterol and protein probes to determine the localisation of PS, the level of membrane electrostatics and the foci of endophilin-induced endocytosis. Overall, I found this manuscript very interesting because of its novelty but also irritating due to several shortcuts and experimental caveats.

Major points:

#1. The experiments with red blood cells are interesting as regards to membrane surface. However, I had difficulty in understanding the experiments with charged probes (see Figures 3 and S3D). Please explain the sentence: The exogenously added +8-FITC gains access to the cytosol of the ghosts and binds to the cytosolic leaflet of the membrane. How does this translocation occur? Please also comment the decline in FM4-64 staining. I also noticed that the Grinstein lab developed several probes for the staining of membranes with different levels of electrostatics (+2, +4, +6, +8 probes). It would have been interesting to see how these probes redistribute upon bMCD treatment in various cell lines. Such experiments would nicely complement those performed with LactC2.

#2. When the cytosol becomes hyperosmotic (figure 5B), other effects might be responsible for the reduction of endocytosis. For example, the cytoskeleton underneath the plasma membrane might be perturbed; BAR domains might not be able to electrostatically interact with the plasma membrane...The data are overinterpreted.

#3. The endophilin foci shown in Figure 5B are seldom and have a quite large and heterogeneous size (although the scale in Figure 6Cii is not indicated). Previous reports, notably by the deCamilli labs show foci that are much smaller (close to the diffraction limit) and numerous (see e.g. Dev Cell. 2009 Dec;17(6):811-22 ; Nat Cell Biol. 2010 Sep;12(9):902-8). By EM and super resolution microscopy these foci correspond to tubes that would be compatible with the theoretical curvature of interest here ($R \approx 20$ to 50 nm). However, none of the pictures shown in Figure 6 fit with such a model. The extent of colocalisation is also very weak (notably in panel D). Last, the authors state 'that cholesterol extraction induced a marked increase in the number and size of endophilin foci'. However, their model implies an increase in number but a decrease in size. Isn't a major contraction with the model?

#4. Please compare your model and results with previous in vitro and cellular studies. E.g. Bruckner et al. Biophys. J. 97, 3113-3122 (2009). Rodal et al., Mol. Biol. Cell 10, 961-974 (1999). A. Subtil et al. Proc. Natl. Acad. Sci. U.S.A. 96, 6775-6780 (1999). In these studies, cholesterol removal causes a blockage in endocytosis.

Other points:

#5 Figure 1C. This binding isotherm is performed with increasing concentration of protein at constant lipid concentration (10 μ M). Not sure this is the best condition to assess the effect of cholesterol on the binding properties of GST-LactC2 with regards to PS because the liposomes

become gradually crowded by the protein. In general it is better to work at a constant protein concentration and to increase the amount of liposomes or, instead, to add increasing % of PS in the liposomes.

#6 The protocol for isolating plasma membrane involves the use of polycationic beads (Figure S1): If PS redistributes to other organelles upon methyl- β -cyclodextrin treatment, the plasma membrane should be less trapped by the beads. Please comment and explain. Please give the bar scale in Figure S1.

#7 In Figure 2, please add for panel C, the same horizontal bars (methyl- β -CD 37% or methyl- β -CD 4%) as in panel B, otherwise one cannot know that these bars apply for the two panels.

Reviewer #2 (Remarks to the Author)

The manuscript by Hiram et al. characterises the redistribution of PS in response to cholesterol extraction. The experiments described have been rigorously performed within the limits imposed by the specificity of the biosensors employed. The main problem with the manuscript is over-interpretation of the reported observations. The title is misleading. It actually needs to reflect results. The Abstract is an over-interpretation of the reported results.

Fig1 shows that M β CD treatment leads to the redistribution of PS from the PM to internal membranes of the endocytic system visualised using the C2 domain of Lactadherin as a biosensor for PS. It is of concern to this reviewer that the biosensor may not be specific for PS; in fact panel 1C shows that GST-LactC2 binds with equivalent affinity to liposomes with 0% PS.

Fig2 then goes on to examine the ATP- and temperature-dependence of this redistribution. Minor issue: Fig2A shows FM4-64 staining only in ATP-depleted cells although the text implies that both ATP-replete & -depleted cells are shown. Also, the phrase "are contiguous yet unconnected to" should correctly be "are adjacent but unconnected to". A case is then made for increased negative charge density on the cytosolic surface of the plasma membrane as a consequence of cholesterol extraction and that such an increase provides the basis for curvature generation. Fig3C shows that M β CD treatment of RBC ghosts results in higher probe (+8-FITC) binding. This observation is interpreted by the authors to demonstrate increased charge density - a simpler explanation could be that M β CD treatment permeabilises the ghosts to some extent allowing the probe greater entry to the interior of the ghosts.

In Fig5D an essential control to include is the ratio for non-M β CD treated high-ionic strength cells (and similarly the ratio for non-M β CD treated distended cells in Fig5G). My concern is that if (as in Fig1B for non-M β CD treated isotonic cells) the value is near 2.5 then the redistribution for PS would be quite robust under the conditions of Fig5D and 5G - a conclusion different from that reached by the authors.

Fig6A shows the reporter GFP-K-Ras-tail to be present on PM-derived tubules resulting from M β CD treatment. Given that the reporter is present on the inner surface of the PM it is likely to remain on PM-derived tubules regardless of their charge. Supplementary Figs 5A, B are not at all convincing - in fact they suggest that all the species examined are lost from the PM following M β CD treatment.

Endophilin foci shown in Fig6B are not convincing. Furthermore, if the authors are trying to suggest that the redistribution of PS is endophilin-mediated then this claim should be validated by knock-down of endophilin.

Although the data presented in this manuscript are of high quality the interpretations are unjustified and over-reaching; a modicum of restraint is required in the re-writing of the

manuscript in order to make it suitable for publication.

Reviewer #3 (Remarks to the Author)

Hirama et al. report on the potential role of anionic phospholipids in driving endocytic plasma membrane invaginations in cells. The authors show that cellular depletion of cholesterol predicted to shield negatively charged lipids in the plasma membrane induces the redistribution of phosphatidylserine (PS) from the surface to endosomes. Application of a fluorescently labelled charged peptide used as charge sensor suggests that cholesterol removal acts by increasing the surface charge density of the plasma membrane. Theoretical modelling further indicates that such a change in surface charge density is sufficient to induce spontaneous curvature due to electrostatic repulsion between PS headgroups. Consistent with this elevation of cytosolic ionic strength or elevation of plasma membrane tension are shown to reduce PS internalization. Finally, it is shown that PS internalization occurs by plasma membrane tubules that contain the BAR domain protein endophilin but are devoid of charged lipids such as PI4,5P2 or PI4P. Based on the data the authors propose that localized increases in charge density may drive endocytosis under certain physiological conditions.

The concept that alterations in surface charge, in particular local increase of charged lipids in the cytoplasmic leaflet of the membrane may drive membrane deformation, e.g. during endophilin-mediated endocytosis is interesting in principle and may warrant publication in a prominent journal, if proven to be of physiological relevance. In the present format the Ms is somewhat preliminary with a number of questions remaining unanswered. These pertain in particular to the question whether the authors can prove their case by demonstrating the use of such a charge-based mechanism for membrane deformation and endocytosis in cells.

1) As alluded to above the key question is whether or not charge-based mechanisms indeed operate in living cells under physiological conditions. Given the prominent association of endophilin with PS-tubules in cholesterol-depleted cells, clathrin independent endocytosis of receptors via endophilin would seem a good candidate. What is the effect of altering ionic strength and of shielding membrane PS by sphingosine and/ or overexpression of LactC2 on endophilin-mediated endocytosis or uptake of Shiga toxin? Do endophilin-covered PS-tubules contain cargo internalized via endophilin-based CIE including Shiga toxin?

2) Related to the above: What is the effect of KD of endophilins or of other BAR domain proteins on the ability of cells to form PS-containing tubules and/ or to internalize PS? Can one rule out that the formation of endophilin-based tubules simply reflects a compensatory mechanism induced by cholesterol depletion rather than representing an endocytic intermediate for PS tubules en route to endosomes?

3) No characterization is presented as to other possible changes in plasma membrane lipid composition induced by cholesterol depletion. Do other charged lipids such as PI4P, PI4,5P2, or PA also redistribute to endosomes? In this context it seems puzzling that the PS-tubules shown in Fig. 6 appear to be devoid of PI4P and PI4,5P2 as one would expect that at least PI4,5P2 associates with cholesterol enriched domains. How can this be explained?

4) The authors show that increasing cytosolic ionic strength counteracts PS internalization. Does lowering of cytosolic ionic strength conversely induce PS tubules and endocytosis?

5) Much of the data represented use complex cellular models. What is the effect of PS on membrane tubulation via endophilin in a simple liposome-based model? Is endophilin under these conditions capable of inducing membrane fission? Does fission of PS-tubules in cells depend on dynamin activity as expected if this pathway was related to endophilin-mediated endocytosis?

Reviewer #1

This manuscript presents a novel hypothesis for the simultaneous presence of the anionic lipids phosphatidylserine and cholesterol in the inner leaflet of the plasma membrane: cholesterol would attenuate the lateral repulsion between PS molecules, which in the absence of phosphatidylserine have a tendency to curve the plasma membrane inward. This effect can be advantageous for endocytosis. The manuscript includes a physical model to estimate the effect of cholesterol removal on PS induced membrane curvature and various cellular assays. They combine cyclodextrin to remove cholesterol and protein probes to determine the localisation of PS, the level of membrane electrostatics and the foci of endophilin-induced endocytosis. Overall, I found this manuscript very interesting because of its novelty but also irritating due to several shortcuts and experimental caveats.

Major points:

#1. The experiments with red blood cells are interesting as regards to membrane surface. However, I had difficulty in understanding the experiments with charged probes (see Figures 3 and S3D). Please explain the sentence: The exogenously added +8-FITC gains access to the cytosol of the ghosts and binds to the cytosolic leaflet of the membrane. How does this translocation occur? Please also comment the decline in FM4-64 staining. I also noticed that the Grinstein lab developed several probes for the staining of membranes with different levels of electrostatics (+2, +4, +6, +8 probes). It would have been interesting to see how these probes redistribute upon bMCD treatment in various cell lines. Such experiments would nicely complement those performed with LactC2.

Thank you for your comments. We apologize for the paucity of detail in the initial submission; due to space restrictions, we did not fully elaborate on the rationale of some of the experiments. We have tried to do so in the revised version. Specifically:

-In Figure 3 & S3D we used red cell ghosts, which are prepared by hypotonic lysis (Schwoch G, Passow H. (1973) *Mol Cell Biochem.* 2(2):197-218; Johnson RM, Kirkwood DH.

Biochim Biophys Acta 1978, 509(1):58-66). The swelling caused by the hypotonic treatment creates pores that allow the efflux of hemoglobin. Under the conditions that we used (5 mM phosphate buffer, 4°C) the pores fail to reseal, allowing virtually complete loss of hemoglobin, hence the designation of the resulting ghosts as “white”, to distinguish them from “pink”, resealed ghosts that are prepared by a different procedure. The large pores that persist in white ghosts enable access by the +8-FITC peptide probe, which is considerably smaller (≈ 2 kDa) than

hemoglobin (≈ 65 kDa). Once it enters the ghost, the probe accumulates as it binds the negatively charged inner leaflet of the membrane, which retains its asymmetric lipid composition. To validate and illustrate the presence of large pores in the membrane of the white ghosts, we have included a new panel, Figure 3E, showing that FITC-dextran of 10 kDa (significantly larger than the +8-FITC probe) enters the ghosts freely and rapidly, but does not associate with the inner leaflet of the membrane. Additionally, we now include panel 3F to show that the +8-FITC peptide equilibrates rapidly and that the levels remain steady in both control ghosts and in the m β CD-treated ghosts for several minutes. Together, these results demonstrate that the differences in fluorescence intensity cannot be attributed to differential accessibility of the probe to the cytoplasmic side of the RBC ghosts. The new information intended to clarify the nature and properties of the ghosts is now described on pages 7-8 of the revised version.

-The decline in FM4-64 was caused by its extraction from the membrane by m β CD. That m β CD can bind and extract FM4-64 from cellular membranes had been reported earlier (see *Kay AR et al. Neuron 1999, PMID 10624945* and *J Neurosci. 2010 Dason JS et al, PMID 21106824* for further details). This is now explained on page 7 of the revised manuscript.

-As suggested by the reviewer we have examined the impact of cholesterol extraction on the localization of the +8, +6 and +4 surface charge probes. These results are now included in the new Supplemental Figure 5 and discussed on page 11 of the revised text. Briefly, the +6 probe –which binds to both the PM and endosomes in control cells– is redistributed to endomembranes concomitant with the internalization of PtdSer. The redistribution is less apparent with the other probes: the +8 probe, which is more sensitive to the charge density of polyanions, remains predominantly associated with the plasma membrane due to the retention of PtdIns(4,5)P₂ and PtdIns4P. The +4 probe associates poorly with the plasmalemma in untreated cells, making its redistribution difficult to detect.

#2. When the cytosol becomes hyperosmotic (figure 5B), other effects might be responsible for the reduction of endocytosis. For example, the cytoskeleton underneath the plasma membrane might be perturbed; BAR domains might not be able to electrostatically interact with the plasma membrane...The data are overinterpreted.

-We agree with the reviewer that the increase in ionic strength will impact the cell in a variety of ways. We have toned down our conclusion and alerted the readers on page 10 that alternative interpretations should not be ruled out. Ultimately, this experiment is only one of several experiments and theoretical calculations that support our conclusions.

#3. The endophilin foci shown in Figure 5B are seldom and have a quite large and heterogeneous size (although the scale in Figure 6Cii is not indicated). Previous reports, notably by the de Camilli lab show foci that are much smaller (close to the diffraction limit) and numerous (see e.g. Dev Cell. 2009 Dec;17(6):811-22 ; Nat Cell Biol. 2010 Sep;12(9):902-8). By EM and super resolution microscopy these foci correspond to tubes that would be compatible with the theoretical curvature of interest here ($R \approx 20$ to 50 nm). However, none of the pictures shown in Figure 6 fit with such a model. The extent of colocalisation is also very weak (notably in plane I D). Last, the authors state 'that cholesterol extraction induced a marked increase in the number and size of endophilin foci'. However, their model implies an increase in number but a decrease in size. Isn't a major contraction with the model?

To address the reviewer's concern about the size and paucity of endophilin foci in our cells, we performed new experiments using super-resolution microscopy to better detect the discrete endophilin puncta. Using structured illumination microscopy (SIM), we now appreciate the presence of a considerable number of smaller foci that were not readily apparent by wide-field or even confocal microscopy (see new Figure 7 and Supplemental Video 1). We also include better images to document the co-localization of endophilin with synaptojanin and with the PtdSer-enriched tubules.

The super-resolution images can also explain why the endophilin puncta are seemingly bigger after cholesterol extraction. Unlike the small puncta seen in untreated cells, the larger endophilin-rich foci seen following cholesterol extraction had irregular shapes, often appearing as multi-lobed structures with tubules emerging from some of them (Figure 7B and Supplemental Movies 1 and 2). These most likely represent serpentine (coiled) accumulations of tubules underneath the membrane, as observed previously by electron microscopy in cholesterol-depleted cells (see image below, reprinted from Shen, H. *et al.* (2014) *Nat. Cell Biol.* **16**, 652–62). Our SIM images are consistent with this array of tubules and, importantly, the diameter of the tubules in the coiled structures reported by EM by Shen *et al.* are in good accord with our theoretical calculations.

COS-7 cells treated with m β CD for ~10 min.

Thank you for alerting us to the omission of the scale bar, we have included it in the revised figure.

#4. Please compare your model and results with previous in vitro and cellular studies. E.g. Bruckner et al. Biophys. J. 97, 3113-3122 (2009). Rodal et al., Mol. Biol. Cell 10, 961-974 (1999). A. Subtil et al. Proc. Natl. Acad. Sci. U.S.A. 96, 6775-6780 (1999). In these studies, cholesterol removal causes a blockage in endocytosis.

The reviewer brings up three important studies on the properties of cholesterol and its role in endocytosis. The Bruckner et al. study suggests that the ability of cholesterol to spontaneously flip-flop across the membrane serves as a buffer, tending to reduce bending energies and the generation of curvature. Indeed, the ability of cholesterol to flip-flop rapidly and spontaneously explains why extracellular m β CD efficiently depletes cholesterol from both leaflets of the PM. In our system, the removal of cholesterol will not only reduce the spacing between the charged headgroups of PtdSer and the phosphoinositides but, as implied by the findings of Bruckner et al, will also reduce the buffering effect normally provided by cholesterol. As a result, the tendency of the membrane to curve as a consequence of electrostatic repulsion will proceed, unimpeded by transmembrane cholesterol redistribution. These considerations have been added to the Discussion (page 15 of the revised text). The work of Bruckner et al is now cited and included in the reference list (new ref. 32). We thank the reviewer for suggesting its inclusion.

The manuscripts of Rodal *et al* (Sandvig's group) and Subtil *et al* (McGraw's group) use similar techniques to show that cholesterol removal by m β CD inhibits clathrin-mediated endocytosis (and not only caveolae, as had been assumed). These studies are by no means in conflict with our observations. They differ from ours not only in the nature of the cargo monitored but, crucially, in the timing of the observations. Both studies depleted cholesterol using 10 mM m β CD for 15–30 min and subsequently investigated clathrin-mediated uptake and/or clathrin pit

dynamics. By contrast, our studies report extensive internalization of PtdSer-rich membranes that occur within 10 min of addition of 10 mM m β CD. These events would not have been apparent to Rodal et al or Subtil et al, who initiated their measurement only afterwards. Moreover, these groups did not measure the fate of PtdSer, as the PtdSer-sensitive LactC2 probe was not available at the time. Thus, the two distinct effects of cholesterol removal can surely co-exist and are not mutually exclusive. These considerations are now discussed on pages 15-16 of the revised Discussion.

Other points:

#5 Figure 1C. This binding isotherm is performed with increasing concentration of protein at constant lipid concentration (10 μ M). Not sure this is the best condition to assess the effect of cholesterol on the binding properties of GST-LactC2 with regards to PS because the liposomes become gradually crowded by the protein. In general it is better to work at a constant protein concentration and to increase the amount of liposomes or, instead, to add increasing % of PS in the liposomes.

The experiments proposed by the reviewer have been performed in the past for LactC2 as well as the related C2 domains of factor V and VII (*Bardelle et al., JBC 1993 and Kay et al., Mol Biol Cell 2012*). The assay we used provides robust, highly reproducible measures of the affinity of LactC2 for PtdSer. As this assay relies on the presence of a FRET acceptor in the liposomes, varying the concentration of liposomes confounds the determinations by changing (increasing) the amount of fluorescence acceptor. Thus, we prefer the assay as performed in this manuscript. In the end, the key finding is that the binding of the probe to PtdSer is indistinguishable whether cholesterol is present or absent, whether under saturating or non-saturating conditions.

#6 The protocol for isolating plasma membrane involves the use of polycationic beads (Figure S1): If PS redistributes to other organelles upon methyl- β -cyclodextrin treatment, the plasma membrane should be less trapped by the beads. Please comment and explain. Please give the bar scale in Figure S1.

The polycationic beads used for PM isolation interact with anionic residues on the outside of intact cells, which are only subsequently sheared off the beads (Supplemental Figure 1). Thus, changes in the charge of the inner leaflet are inconsequential. Note that following binding of the cells and prior to shearing, the unoccupied sites on the beads are blocked by heparin, a polyanion. Additionally, the

values are normalized per protein content, which would account for any possible differences in membrane isolation efficiency.

Thank you for pointing out the absence of the scale bar; it has now been added to Figure S1.

#7 In Figure 2, please add for panel C, the same horizontal bars (mbetaCD 37{degree sign}C or mbetaCD 4{degree sign}C) as in panel B, otherwise one cannot know that these bars apply for the two panels.

Thank you for pointing out this omission. We have corrected this in the revised Figure 2.

Reviewer #2 (Remarks to the Author):

The manuscript by Hirama et al. characterises the redistribution of PS in response to cholesterol extraction. The experiments described have been rigorously performed within the limits imposed by the specificity of the biosensors employed. The main problem with the manuscript is over-interpretation of the reported observations. The title is misleading. It actually needs to reflect results. The Abstract is an over-interpretation of the reported results.

Fig1 shows that MBCD treatment leads to the redistribution of PS from the PM to internal membranes of the endocytic system visualised using the C2 domain of Lactadherin as a biosensor for PS. It is of concern to this reviewer that the biosensor may not be specific for PS; in fact panel 1C shows that GST-LactC2 binds with equivalent affinity to liposomes with 0% PS.

Thank you for your comments. Obviously, binding of Lact-C2 to membranes devoid of PtdSer would be troubling. However, the specificity of Lact-C2 for has been documented extensively (Shi J, Heegaard CW, Rasmussen JT, Gilbert GE. *Biochim Biophys Acta*. 2004;1667(1):82-90; Otzen DE1, Blans K, Wang H, Gilbert GE, Rasmussen JT.

Biochim Biophys Acta. 2012;1818(4):1019-27; Yeung T, Gilbert GE, Shi J, Silvius J, Kapus A, Grinstein S. *Science*. 2008, 11;319(5860); Fairn GD1, Schieber NL, Ariotti N, Murphy S, Kuerschner L, Webb RI, Grinstein S, Parton RG. *J Cell Biol*. 2011;194(2):257-75). The small amount of fluorescence detected in the absence of PtdSer in Fig. 1C is likely background emission, not due to FRET between the probe and the liposomes, as suggested by the fact that it does not change significantly as the concentration of Lact-C2 is varied over a 10-fold concentration range.

Fig2 then goes on to examine the ATP- and temperature-dependence of this redistribution. Minor issue: Fig2A shows FM4-64 staining only in ATP-depleted cells although the text implies that both ATP-replete & -depleted cells are shown.

Also, the phrase "are contiguous yet unconnected to" should correctly be "are adjacent but unconnected to". A case is then made for increased negative charge density on the cytosolic surface of the plasma membrane as a consequence of cholesterol extraction and that such an increase provides the basis for curvature generation.

Thank you for pointing out these two misstatements. We have corrected them on page 6 of the revised manuscript.

Fig3C shows that MβCD treatment of RBC ghosts results in higher probe (+8-FITC) binding. This observation is interpreted by the authors to demonstrate increased charge density - a simpler explanation could be that MβCD treatment permeabilises the ghosts to some extent allowing the probe greater entry to the interior of the ghosts.

A similar question was also raised by reviewer #1. Briefly, the ghosts we used were of the "white" or leaky type, that do not restrict the permeation of proteins, let alone small peptides. For more detail, we copy below the explanation provided earlier:

-In Figure 3 & S3D we used red cell ghosts, which are prepared by hypotonic lysis (Schwoch G, Passow H. (1973) *Mol Cell Biochem.* 2(2):197-218; Johnson RM, Kirkwood DH.

Biochim Biophys Acta 1978;509(1):58-66). The swelling caused by the hypotonic treatment creates pores that allow the efflux of hemoglobin. Under the conditions that we used (5 mM phosphate buffer, 4°C) the pores fail to reseal, allowing virtually complete loss of hemoglobin, hence the designation of the resulting ghosts as "white", to distinguish them from "pink", resealed ghosts that are prepared by a different procedure. The large pores that persist in white ghosts enable ready access by the +8-FITC peptide probe, which is considerably smaller (≈2 kDa) than hemoglobin (≈65 kDa). Once it enters the ghost, the probe accumulates as it binds the negatively charged inner leaflet of the membrane, which retains its asymmetric lipid composition. To validate and illustrate the presence of large pores in the membrane of the white ghosts, we have included a new panel, Figure 3E, showing that FITC-dextran of 10 kDa (significantly larger than the +8-FITC probe) enters the ghosts freely and rapidly, but does not associate with the inner leaflet of the membrane. Additionally, we now include panel 3F to show that the +8-FITC peptide equilibrates rapidly and that the levels remain steady in both control ghosts and in

the m β CD -treated ghosts for several minutes. Together, these results demonstrate that the differences in fluorescence intensity cannot be attributed to differential accessibility of the probe to the cytoplasmic side of the RBC ghosts. The new information intended to clarify the nature and properties of the ghosts is now described on pages 7-8 of the revised version.

In Fig5D an essential control to include is the ratio for non-MBCD treated high-ionic strength cells (and similarly the ratio for non-MBCD treated distended cells in Fig5G). My concern is that if (as in Fig1B for non-MBCD treated isotonic cells) the value is near 2.5 then the redistribution for PS would be quite robust under the conditions of Fig5D and 5G - a conclusion different from that reached by the authors.

Thank you for your comment. These are complex experiments and we debated how best to show the data. We have revised the two graphs, Fig. 5D,H. Increasing the ionic strength of the cytosol tended to decrease the PM:cytoplasm distribution of LactC2 even before cholesterol extraction, while after m β CD extraction the ratio remained unchanged or even increased in the cells with 600 mM NaCl. As for the Figure 5G and H, swelling the cells did not alter the PM:cytoplasm ratio. However, following extraction of cholesterol the relocalization of the LactC2 probe was partly inhibited in the swollen cells, presumably due distention of the plasma membrane.

Fig6A shows the reporter GFP-K-Ras-tail to be present on PM-derived tubules resulting from MBCD treatment. Given that the reporter is present on the inner surface of the PM it is likely to remain on PM-derived tubules regardless of their charge.

While we appreciate the reviewer's comment, we respectfully disagree with his/her conclusion. The association of the charge-sensitive probes with the membrane is rapidly reversible, as can be readily shown using the rapamycin-induced heterodimerization system, or by acutely altering the charge of the membrane. One example is provided by depletion of PtdIns(4,5)P₂, which is accompanied by immediate loss of the K-Ras-tail probe from the plasma membrane (*Heo et al., Science 2006*). Similarly, addition of agents like sphinganine or dibucaine that depress the surface charge also cause acute detachment of the probes from the plasma membrane PLC (*Yeung et al., Science 2006*). Importantly, the R-pre probe detaches from the nascent phagosomal membrane as PtdIns(4,5)P₂ is hydrolyzed by PLC, even prior to phagosome sealing (see Fig. 3H in *Yeung et al., Science 2006*). Thus, we see no reason why the probes would remain on the tubules merely because they were derived from the plasma membrane.

Supplementary Figs 5A, B are not at all convincing - in fact they suggest that all the species examined are lost from the PM following MBCD treatment.

In the revised manuscript, we have removed Supplemental Figure 5 and included new results with clearer images and quantitation in the new figure 6. We apologize for the quality of the previous image.

Endophilin foci shown in Fig6B are not convincing. Furthermore, if the authors are trying to suggest that the redistribution of PS is endophilin-mediated then this claim should be validated by knock-down of endophilin.

Thank you for your suggestions. As requested by the reviewer we performed experiments to knock down of both endophilin A1 and A2 to assess their requirement for the tubulation and scission events. The new results are presented in Fig. 7 of the revised paper. In a nutshell, we found that while endophilin is recruited to the tubules, it is not, in fact, essential for the internalization of the PtdSer-tubules.

Reviewer #1 was similarly concerned about the appearance of the endophilin foci and suggested the use of super-resolution microscopy to better visualize them. We include below the response to Reviewer #1, which addresses also the concerns of this reviewer (#2):

To address the reviewer's concern about the size and paucity of endophilin foci in our cells, we performed new experiments using super-resolution microscopy to better detect the discrete endophilin puncta. Using structured illumination microscopy (SIM), we now appreciate the presence of a considerable number of smaller foci that were not readily apparent by wide-field or even confocal microscopy (see new Figure 7 and Supplemental Video 1). We also include better images to document the co-localization of endophilin with synaptojanin and with the PtdSer-enriched tubules.

The super-resolution images can also explain why the endophilin puncta are seemingly bigger after cholesterol extraction. Unlike the small puncta seen in untreated cells, the larger endophilin-rich foci seen following cholesterol extraction had irregular shapes, often appearing as multi-lobed structures with tubules emerging from some of them (Figure 7B and Supplemental Movies 1 and 2). These most likely represent serpentine (coiled) accumulations of tubules underneath the membrane, as observed previously by electron microscopy in cholesterol-depleted cells (see image below, reprinted from Shen, H. *et al.* (2014) *Nat. Cell Biol.* **16**, 652–62). Our SIM images are consistent with this array of tubules and, importantly, the

diameter of the tubules in the coiled structures reported by EM by Shen *et al.* are in good accord with our theoretical calculations.

COS-7 cells treated with m β CD for ~10 min.

Although the data presented in this manuscript are of high quality the interpretations are unjustified and over-reaching; a modicum of restraint is required in the re-writing of the manuscript in order to make it suitable for publication.

We appreciate your comments. In our excitement to share our findings with the cell biology community, we were perhaps too rash. We have revised the text with this in mind, trying to be more cautious and restrained in the interpretation of our findings.

Reviewer #3 (Remarks to the Author):

Hirama et al. report on the potential role of anionic phospholipids in driving endocytic plasma membrane invaginations in cells. The authors show that cellular depletion of cholesterol predicted to shield negatively charged lipids in the plasma membrane induces the redistribution of phosphatidylserine (PS) from the surface to endosomes. Application of a fluorescently labelled charged peptide used as charge sensor suggests that cholesterol removal acts by increasing the surface charge density of the plasma membrane. Theoretical modelling further indicates that such a change in surface charge density is sufficient to induce spontaneous curvature due to electrostatic repulsion between PS headgroups. Consistent with this elevation of cytosolic ionic strength or elevation of plasma membrane tension are shown to reduce PS internalization, Finally, it is shown that PS internalization occurs by plasma membrane tubules that contain the BAR domain protein endophilin but are devoid of charged lipids such as PI4,5P2 or PI4P. Based on the data the authors propose that localized increases in charge density may drive endocytosis under certain physiological conditions.

The concept that alterations in surface charge, in particular local increase of charged

lipids in the cytoplasmic leaflet of the membrane may drive membrane deformation, e.g. during endophilin-mediated endocytosis is interesting in principle and may warrant publication in a prominent journal, if proven to be of physiological relevance. In the present format the Ms is somewhat preliminary with a number of questions remaining unanswered. These pertain in particular to the question whether the authors can prove their case by demonstrating the use of such a charge-based mechanism for membrane deformation and endocytosis in cells.

1) As alluded to above the key question is whether or not charge-based mechanisms indeed operate in living cells under physiological conditions. Given the prominent association of endophilin with PS-tubules in cholesterol-depleted cells, clathrin independent endocytosis of receptors via endophilin would seem a good candidate. What is the effect of altering ionic strength and of shielding membrane PS by sphingosine and/ or overexpression of LactC2 on endophilin-mediated endocytosis or uptake of Shiga toxin? Do endophilin-covered PS-tubules contain cargo internalized via endophilin-based CIE including Shiga toxin?

We have performed quite a number of new experiments to address the interesting suggestions made by the reviewer to enhance our manuscript further:

-The recruitment of endophilin and the tubular nature of the structures formed by depletion of cholesterol are indeed reminiscent of the endocytosis initiated by Shiga toxin. We analyzed if tubules induced by this family of toxins are PtdSer-positive and whether their formation requires negative charge in the cytoplasmic leaflet. To this end, HeLa cells expressing GFP-LactC2 were treated with Alexa568-labeled verotoxin, a Shiga-like toxin that similarly binds to glycolipids and induces tubulation and endocytosis. As shown in the Figure below, panel A, verotoxin does stimulate the formation of PtdSer-containing tubular carriers.

-If negative charge is necessary for verotoxin-mediated endocytosis, we hypothesized that the addition of a positively charged amphipathic molecule – expected to reduce the negative potential of the inner leaflet– would block its internalization. We initially confirmed that addition of sphingosine to HeLa cells causes release of the R-Pre surface charge probe, which redistributes rapidly to endomembranes (see panel B below). Furthermore, we found that sphingosine displaces endophilin2 from m β CD-induced foci (panel C), consistent with reduced anionic surface charge. Most importantly, pretreatment with sphingosine prevented the membrane tubulation and internalization caused by verotoxin (panel D below).

-We tried to detect whether cholesterol depletion induced verotoxin internalization. This required the use of conditions where verotoxin would not itself cause tubulation. However, we found that even the lowest visible concentrations that we tested produced internalization, confounding the effects of m β CD. We could therefore not establish unambiguously whether

verotoxin was present in the m β CD-induced PtdSer-rich tubules (as opposed to inducing the tubules by itself).

2) Related to the above: What is the effect of KD of endophilins or of other BAR domain proteins on the ability of cells to form PS-containing tubules and/ or to internalize PS? Can one rule out that the formation of endophilin-based tubules simply reflects a compensatory mechanism induced by cholesterol depletion rather than representing an endocytic intermediate for PS tubules en route to endosomes

We agree with the reviewer that the data presented in the original version could not distinguish whether endophilin was causing the tubulation, acting as a compensatory mechanism or merely serving as a curvature sensor. In view of the recent discovery of a fast mode of endophilin-mediated endocytosis, we initially speculated (admittedly without direct evidence to support it) that endophilin was supporting the tubulation. To examine this more formally, after appreciating the reviewers' concern, we silenced endophilin A1 and A2 in HeLa cells separately and in combination. As shown in the new Figure 7E and F, we could achieve nearly complete knock-down of the mRNA for each isoform, yet this did not impact the tubulation and redistribution of PtdSer. Thus, the data suggest that endophilin is a marker and possibly a compensatory response to tubulation, but is not essential for the process.

3) No characterization is presented as to other possible changes in plasma membrane lipid composition induced by cholesterol depletion. Do other charged lipids such as PI4P, PI4,5P2, or PA also redistribute to endosomes? In this context it seems puzzling that the PS-tubules shown in Fig. 6 appear to be devoid of PI4P and PI4,5P2 as one would expect that at least PI4,5P2 associates with cholesterol enriched domains. How can this be explained?

Thank you for your comment; this is an important consideration. We performed additional experiments and now show in Figure 6 that the levels of PtdIns4P and PtdIns(4,5)P₂ in the PM are not discernibly affected by cholesterol extraction. The phosphoinositides are, however, depleted from the tubules. This is most likely due to the accumulation of synaptojanin at their base, where they emerge from the membrane and where synaptojanin was found to co-localize with endophilin (see new Figure 6). As synaptojanin possesses dual 5-phosphatase and 4-phosphatase activity, its recruitment could effectively deplete PtdIns4P and PtdIns(4,5)P₂ from the emerging tubules.

Despite some early suggestions that PtdIns(4,5)P₂ associates with cholesterol, subsequent literature has cast doubt on this notion. The bulk of the PtdIns and PtdIns(4,5)P₂ in the plasma membrane contains stearate (18:0) and arachidonate

(20:4) at the *sn*-1 and *sn*-2 positions, respectively. The high degree of unsaturation at the *sn*-2 position is not conducive to interaction with cholesterol. While the existence of a fraction of di-saturated PtdIns(4,5)P₂ has not been formally ruled out, this would be a minor component of the membrane; the behavior of this putative subpopulation under circumstances where cholesterol is extracted is hard to predict.

4) The authors show that increasing cytosolic ionic strength counteracts PS internalization. Does lowering of cytosolic ionic strength conversely induce PS tubules and endocytosis?

Under the conditions we have tested, we have not noticed any tubules in our cells due to the lowering of cytosolic ionic strength. Because lowering the ionic strength by diluting the medium is accompanied by osmotic swelling, we only applied relatively modest changes. However, the reviewer's question prompted us to read potentially relevant literature and we found that previous work, such as that of van der Wijk et al., (*J. Biol. Chem.*, 2003) demonstrated a nearly ≈ 100 -fold increase in the endocytosis of fluorescent dextran following hypotonicity-induced swelling. While the precise signaling events driving this endocytosis are currently unknown, roles for myosin light-chain kinase and Src kinase have been postulated in epithelial cells (e.g. Barford et al., *Mol Biol Cell* 2011).

5) Much of the data represented use complex cellular models. What is the effect of PS on membrane tubulation via endophilin in a simple liposome-based model? Is endophilin under these conditions capable of inducing membrane fission? Does fission of PS-tubules in cells depend on dynamin activity as expected if this pathway was related to endophilin-mediated endocytosis?

These are also very insightful questions. Indeed, it was shown previously that recombinant endophilin can tubulate PtdSer-rich, PIP-deficient, liposomes (*Farsad et al. JCB* 2001; PMID 11604418). However, from the data available in that report it is unclear whether significant fission occurred.

Prompted by the reviewer's question, we tested the role of dynamin. We have found that, under conditions where transferrin uptake was obliterated, inhibition of dynamin with Dyno4A did not prevent the internalization of PtdSer following the extraction of cholesterol. This is consistent with the finding that ATP depletion –and by extension GTP depletion also– does not inhibit the internalization. The observation that the endocytosis induced by cholesterol depletion does not require dynamin is now described and discussed on page 15 of the revised text.

Reviewers' Comments:

Reviewer #1 (Remarks to the Author)

This is a carefully prepared revision. The authors have addressed my points by both textual changes and new experiments. I recommend publication. Please note, however, that I could not visualize movies 1 and 2 (these files are seen by my computer as 'corrupted')

Reviewer #3 (Remarks to the Author)

In their revised Ms Hirama et al have added some extra data to support their proposal that elevated concentrations of charged lipids, in particular PS, induced by depletion of cholesterol can induced endocytic tubulation events in cells that result in PS removal and redistribution to the endosomal system.

I have to admit that I remain unconvinced of the physiological relevance of the observations made. I don't doubt any of the data I still miss clear-cut causal evidence that the data go beyond a peculiar observation induced by beta-methylcyclodextrin (bmCD) rather than representing a key step in endocytosis of any kind. That said, many of the data are merely correlative and no direct manipulation of PS content has been performed, which allows for a number of alternative explanations that in my view preclude publication in a top-class journal such as Nature Communications.

1. While bmCD can deplete cholesterol the question remains what the global effects of such treatment on general cell physiology might be as bmCD can sequester a variety of other molecules in addition to cholesterol. Although the authors do not detect significant alterations of PM PI4P and PI4,5P2 content it remains possible that other lipids such as glycolipids, which themselves have been implicated in clathrin-independent endocytosis and membrane tubulation (e.g. of toxins but also of endogenous cargo such as galectins), or PA are also disturbed. Hence, more direct manipulation of cellular or even better plasma membrane PS content is needed to support the central claims of the paper.

2. As said above it remains unclear if and to what extent the observations are relevant for clathrin- and possibly dynamin-independent endocytosis. The presented data for reviewers regarding verotoxin seem inconclusive and have not been incorporated into the Ms, presumably because it may have been difficult to distinguish toxin effects on membrane shape from those induced by altered cytoplasmic leaflet PS content. I suggest that the authors invest additional efforts into this important issue. I suggest to study Shiga rather than verotoxin as its endocytic pathway is better characterized and different mutants are available that may allow to distinguish PS from toxin-induced lipid clustering effects. If the authors' hypothesis is correct then alterations in PS content/ clustering should synergize with effects of Shiga on Gb3 clustering. This proposal is testable in my view. Does lowering of cellular PS content, e.g. using RNAi reduce Shiga toxin endocytosis?

3. The data regarding manipulation of ionic strength are not entirely convincing nor are they consistent. While elevated ionic strength indeed appears to prevent PS internalization, reduced ionic strength (although interpreted differently by the authors) yields similar though less complete effects. This is at odds with the theoretical model and with the studies cited in the response to reviewers (Wijk et al JBC 2003). Clearly, other approaches to selectively target PS within the cytoplasmic leaflet are needed to investigate this point.

Minor points (addressable by text changes)

4. It is unclear to me to what degree the different charge probes recognize PIPs vs PS vs PA. In the absence of a clear characterization of the +4, +6, and +8 charge probes their behavior needs

to be interpreted with caution.

5. The absence of PIP4P or PI4,5P2 probes should not be taken as formal proof of the absence of these lipids from the LactC2 positive tubules as proteins that bind to these lipids may not allow access to PI lipid probes. This caveat should at least be acknowledged in the results and discussion sections of the Ms.

Reviewer #4 (Remarks to the Author)

The authors have carefully addressed the concerns of the reviewer. New experiments have been added to the manuscript which is now much more convincing and should be accepted for publication.

Reviewer #3 (Remarks to the Author):

In their revised Ms Hirama et al have added some extra data to support their proposal that elevated concentrations of charged lipids, in particular PS, induced by depletion of cholesterol can induced endocytic tubulation events in cells that result in PS removal and redistribution to the endosomal system.

I have to admit that I remain unconvinced of the physiological relevance of the observations made. I don't doubt any of the data I still miss clear-cut causal evidence that the data go beyond a peculiar observation induced by beta-methylcyclodextrin (bmCD) rather than representing a key step in endocytosis of any kind. That said, many of the data are merely correlative and no direct manipulation of PS content has been performed, which allows for a number of alternative explanations that in my view preclude publication in a top-class journal such as Nature Communications.

1. While bmCD can deplete cholesterol the question remains what the global effects of such treatment on general cell physiology might be as bmCD can sequester a variety of other molecules in addition to cholesterol. Although the authors do not detect significant alterations of PM PI4P and PI4,5P2 content it remains possible that other lipids such as glycolipids, which themselves have been implicated in clathrin-independent endocytosis and membrane tubulation (e.g. of toxins but also of endogenous cargo such as galectins), or PA are also disturbed. Hence, more direct manipulation of cellular or even better plasma membrane PS content is needed to support the central claims of the paper.

We agree that showing the effect of charge density on membrane curvature and endocytosis is indeed important. To address this concern, we have implemented an entirely different means of increasing the negative charge of the inner leaflet of the plasma membrane (PM). PtdSer synthase 1 and 2 undergo product feedback inhibition. Thus, merely over-expressing the ER-resident synthases is insufficient to increase the levels of plasmalemmal PtdSer. The genetic disorder, Lenz-Majewski syndrome is caused by mutations in PtdSer Synthase 1 which result in loss of this feedback inhibition. However, expression of the mutant PtdSer synthase 1 leads to the accumulation of PtdSer in the ER with no impact on the plasmalemma (T. Balla personal communications and Sohn et al., PNAS 2016). As an alternative approach, in the revised manuscript we decided to increase the PtdSer content of the PM by adding exogenous PtdSer to the medium, as liposomes. Several studies have demonstrated that PtdSer is rapidly incorporated into the outer leaflet of the PM and flipped to the cytosolic leaflet, where it will increase the anionic charge. When incubating cells under these conditions for 15 min, we find that the LactC2 probe relocalizes to endosomal structures, consistent with the findings described earlier using m β CD experiments. Images and quantitation of these experiments are now included as new panels D and E in the revised version of the pre-existing Figure 4.

Furthermore, the addition of exogenous PtdSer to cells also led to an increase in the number of Synaptojanin-Endophilin positive foci, consistent with the notion that curvature and tubulation were induced. This result is now included in the updated Figure 6 as the new panel G. Together with the previous observations, these new results demonstrate that increasing cytosolic leaflet

PtdSer stimulates the endocytosis of PtdSer-containing membranes and leads to increased endophilin and synaptojanin recruitment to the PM.

2. As said above it remains unclear if and to what extent the observations are relevant for clathrin- and possibly dynamin-independent endocytosis. The presented data for reviewers regarding verotoxin seem inconclusive and have not been incorporated into the Ms, presumably because it may have been difficult to distinguish toxin effects on membrane shape from those induced by altered cytoplasmic leaflet PS content. I suggest that the authors invest additional efforts into this important issue. I suggest to study Shiga rather than verotoxin as its endocytic pathway is better characterized and different mutants are available that may allow to distinguish PS from toxin-induced lipid clustering effects. If the authors' hypothesis is correct then alterations in PS content/ clustering should synergize with effects of Shiga on Gb3 clustering. This proposal is testable in my view. Does lowering of cellular PS content, e.g. using RNAi reduce Shiga toxin endocytosis?

Verotoxin, like the closely related Shiga toxin and other AB type bacterial toxins, binds and clusters lipids in the exofacial leaflet of the PM and in turn induces deformations of the PM that result in a tubular endocytosis. It is unclear to what extent a negative surface charge on the inner leaflet of the PM is required to support the internalization. To address this further, we first used sphingosine, a positively charged lipid, to neutralize the charge. As highlighted in Supplemental Figure 6A, the addition of sphingosine completely abolished the internalization of Cy3 labeled verotoxin, with no impact on the binding to Gb3.

PtdSer is essential for the viability of mice and, to date, no one has reported complete silencing of the two PtdSer synthase isoforms. That said, we do have a CHO cell line, named PSB-2, that has a $\approx 95\%$ reduction in the total PtdSer synthase activity which translates into a $\approx 75\text{-}80\%$ reduction in total cellular PtdSer, where the role of PtdSer in toxin internalization could in principle be analyzed. Normally, CHO cells do not produce Gb3 and are hence incapable of binding Shiga or verotoxin (1). However, we were able to induce the generation of Gb3 in CHO and PSB-2 cells by transfecting a plasmid encoding the enzyme alpha 1,4-galactosyltransferase (Gb3 synthase), as reported earlier(1). We verified that, while verotoxin is unable to bind to untransfected CHO or PSB-2 cells, it does bind to cells heterologously expressing the Gb3 synthase, consistent with previous findings (ref). Using such Gb3 producing CHO and PSB-2 cells we examined the role of PtdSer in verotoxin-mediated endocytosis and found that a $\sim 75\%$ reduction in PtdSer did not impair endocytosis.

Unfortunately, this experiment is complicated by compensatory changes in the lipidome when cells are depleted chronically of PtdSer. Similar compensation is likely to occur in the course of siRNA mediated synthase depletion suggested by the reviewer. In this regard, it is important to note that the anionic lipid, phosphatidylinositol, replaces much of the PtdSer loss in PSB-2 cells (2). Phosphatidylinositol is also the precursor for the less abundant but higher valency possessing phosphoinositides. Our theoretical calculations suggest that any abundant anionic phospholipid, not just PtdSer, could induce endocytosis.. Thus, the substitution of PtdSer with PtdIns (and potentially phosphoinositides) could explain why PSB-2 cells internalize

verotoxin, while charge neutralization abolishes it. In support of this notion, we find that the R-pre probe remains associated with the plasma membrane in PSB-2 cells (see figure), arguing for maintenance of negative surface charge.

GFP-R-Pre in PSB-2 cells.

While these findings may be of potential interest to some readers, they are not critical to understanding the primary findings of the paper. Therefore, they are included as a new Supplemental Figure 6.

3. The data regarding manipulation of ionic strength are not entirely convincing nor are they consistent. While elevated ionic strength indeed appears to prevent PS internalization, reduced ionic strength (although interpreted differently by the authors) yields similar though less complete effects. This is at odds with the theoretical model and with the studies cited in the response to reviewers (Wijk et al. JBC 2003). Clearly, other approaches to selectively target PS within the cytoplasmic leaflet are needed to investigate this point.

The Wijk et al manuscript uses different conditions to activate a signaling response. In these experiments, the authors induced swelling of the cells to promote and study a cell volume regulatory response (regulatory volume decrease); this compensatory shrinking results from loss of KCl and facilitates endocytosis during the volume regulatory phase. The reviewer may not have appreciated that we purposefully maintained the intra- and extracellular KCl product constant in order to prevent regulatory shrinking. Under these conditions, membrane stretching persists and clearly endocytosis is inhibited. The example was included to demonstrate that the reviewer's hypothesis has been reported in the literature, although, we do not believe it to be relevant to these experiments. While imposition of hypotonicity will indeed alter other cellular parameters, increased lateral membrane tension will certainly be a major component and an obstacle for the formation of spontaneous membrane curvature (ref). In the end, we believe the two experiments described above help clarify the cellular response we are describing.

Minor points (addressable by text changes)

4. It is unclear to me to what degree the different charge probes recognize PIPs vs PS vs PA. In

the absence of a clear characterization of the +4, +6, and +8 charge probes their behavior needs to be interpreted with caution.

We have added references describing the previous characterization of the probes. Because they are simple polycationic sequences, there is no reason to believe that they would show selectivity for specific lipids.

5. The absence of PIP4P or PI4,5P₂ probes should not be taken as formal proof of the absence of these lipids from the LactC2 positive tubules as proteins that bind to these lipids may not allow access to PI lipid probes. This caveat should at least be acknowledged in the results and discussion sections of the Ms.

Thanks you for the comment, we fully agree with the statement. We always try to remind readers that these types of probes only bind accessible ligands. We have added to the text the caveat that the PI4P and PI4,5P₂ probes only bind available (exposed) lipids.

1. Keusch, J. J., Manzella, S. M., Nyame, K. A., Cummings, R. D., and Baenziger, J. U. (2000) Cloning of Gb3 synthase, the key enzyme in globo-series glycosphingolipid synthesis, predicts a family of alpha 1, 4-glycosyltransferases conserved in plants, insects, and mammals. *The Journal of biological chemistry* **275**, 25315-25321
2. Saito, K., Nishijima, M., and Kuge, O. (1998) Genetic evidence that phosphatidylserine synthase II catalyzes the conversion of phosphatidylethanolamine to phosphatidylserine in Chinese hamster ovary cells. *The Journal of biological chemistry* **273**, 17199-17205

Reviewers' Comments:

Reviewer #3:

Remarks to the Author:

I very much appreciate the additional efforts and explanations that Hiram et al have provided. The new data on PS loading of cells shown in Fig.4 certainly support the proposed model although they do not rigorously prove it in my view. As verotoxin internalization data in mutant cells partially depleted of PS persists, while compensatory changes in PI and PIP content in these cells are likely to occur, I suggest that these data (shown in Fig. S6C,D) be removed from the final version of the Ms and leave this for the discussion.

REVIEWERS' COMMENTS:

Reviewer #3 (Remarks to the Author):

I very much appreciate the additional efforts and explanations that Hiramama et al. have provided. The new data on PS loading of cells shown in Fig.4 certainly support the proposed model although they do not rigorously prove it in my view. As verotoxin internalization data in mutant cells partially depleted of PS persists, while compensatory changes in PI and PIP content in these cells are likely to occur, I suggest that these data (shown in Fig. S6C,D) be removed from the final version of the Ms and leave this for the discussion.

Thank-you for your suggestion. However, as the policy of the journal is not to use “data not shown” we have opted to keep the supplementary figure 6C and D in the final version.